# FasMe: Fast and Sample-efficient Meta Estimator for Precision Matrix Learning in Small Sample Settings

**Xiao Tan**[1,3]    **Yiqin Wang**[1]    **Yangyang Shen**[1]    **Dian Shen**[1]
**Meng Wang**[2]    **Peibo Duan**[3]    **Beilun Wang**[1,*]

[1]School of Computer Science and Engineering, Southeast University, Nanjing, China
[2]College of Design and Innovation, Tongji University, Shanghai, China
[3]Department of Data Science & AI, Monash University, Melbourne, Australia
{xtan, 213200449, seu_syy, dshen, beilun}@seu.edu.cn
mengwangtj@tongji.edu.cn   peibo.duan@monash.edu

## Abstract

Precision matrix estimation is a ubiquitous task featuring numerous applications such as rare disease diagnosis and neural connectivity exploration. However, this task becomes challenging in small sample settings, where the number of samples is significantly less than the number of dimensions, leading to unreliable estimates. Previous approaches either fail to perform well in small sample settings or suffer from inefficient estimation processes, even when incorporating meta-learning techniques. To this end, we propose a novel approach FasMe for Fast and Sample-efficient Meta Precision Matrix Learning, which first extracts meta-knowledge through a multi-task learning diagram. Then, meta-knowledge constraints are applied using a maximum determinant matrix completion algorithm for the novel task. As a result, we reduce the sample size requirements to $O(\log p/K)$ per meta-training task and $O(\log |\mathcal{G}|)$ for the meta-testing task. Moreover, the hereby proposed model only needs $O(p \log \epsilon^{-1})$ time and $O(p)$ memory for converging to an $\epsilon$-accurate solution. On multiple synthetic and biomedical datasets, FasMe is at least ten times faster than the four baselines while promoting prediction accuracy in small sample settings.

## 1 Introduction

Precision matrix estimation aims to exploit conditional dependency relationships between random variables, which plays a vital role in high-dimensional statistical learning with a wide range of applications in areas such as genetics [1], neuroscience [2], and social networks. For instance, researchers are examining the gene network derived from the gene expression data of patients to investigate a rare disease. However, in cases involving rare diseases, the analysis is hindered by the high dimensionality of genetic datasets, where the number of features significantly exceeds the number of samples. For example, the Cholangiocarcinoma dataset in TCGA contains only 51 samples but thousands of genes. In this context, we define a "small sample setting" as a scenario in which the number of samples is less than one-tenth of the number of dimensions. Although inferring such relationship graphs in small sample settings is challenging, it can provide valuable insights into the genetic characteristics underlying this disease and its etiology.

Precision matrix estimator inherently has the capability to handle datasets where the number of dimensions $p$ is equal to or greater than the number of samples $n$ [3]. However, this capability has its

---

\* Corresponding Author.

38th Conference on Neural Information Processing Systems (NeurIPS 2024).

limits, and maintaining effectiveness requires keeping the sample size at a certain level. [4] provides the lower bound of the sample size as $\Omega(d^2 \log p)$ for well-performed graph recovery under the sub-Gaussian assumption, where $d$ is the maximum node degree of the graph. Consequently, when the sample size is reduced to the aforementioned "small sample" levels, the sample size falls short of this theoretical lower bound with a relatively high probability due to the uncertainty of $d$. This indicates insufficient data support for most methods, like GLasso [5] (detailed in Background), to recover the graph structure effectively, as shown in Figure 1. To address this issue, some methods use multi-task learning, integrating heterogeneous datasets to reduce the sample size bound. For example, the regularized methods [6] and [7] show that each task only needs $O(\log K + \log p)$ and $O(K + \log p)$ samples, respectively, for proper estimation, and $K$ denotes the number of tasks. Although these methods are more capable of handling small sample problems, they concurrently introduce significant computational burdens. Specifically, integrating a new task necessitates repeated joint retraining, thereby resulting in substantial computational inefficiencies.

To address these challenges, meta-learning emerges as a promising paradigm, as it equips models to efficiently learn a new task using minimal data through the application of meta-knowledge. Such methods are particularly attractive for precision matrix estimation in small sample settings, as they diminish the reliance on extensive data and reduce computational demands. For example, state-of-the-art approaches from the class of Model-Agnostic Meta-Learning (MAML) [8] have achieved favorable performance in many meta-learning tasks. The strength of MAML lies in its ability to quickly adapt to new tasks (meta-testing dataset) with fewer samples by optimizing initial parameters using data from related tasks (meta-training dataset).

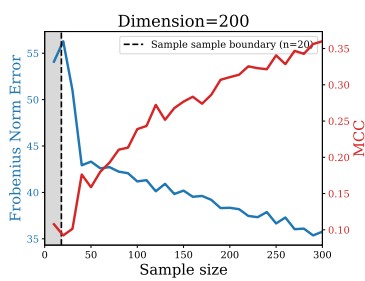

Figure 1: Frobenius norm error ($\downarrow$) and Matthews Correlation Coefficient (MCC, $\uparrow$) V.S. sample size using GLasso. We fix $p = 200$ and vary $n$ from $10$ to $400$ with step $10$. The model performance drops sharply around $n = 20$, indicating its inadequacy for small sample.

However, the design of MAML often emphasizes sample efficiency and fast adaptation, typically without a comprehensive and robust theoretical guarantee for its performance. While empirical improvements are sufficient for many neural network applications, they may lead to incorrect predictions when applied to precision matrix estimation, where precise and reliable results are crucial. Such empirical-only strategies typically rely on extensive training data. However, in precision matrix estimation, the available data is often significantly limited. For example, the dropout technique [9], which lacks a strict theoretical foundation, is frequently used in neural network training with millions of samples. In contrast, in precision matrix estimation, the volume of available data often falls short of the number of feature dimensions. Furthermore, MAML-based methods adapt to new tasks by performing gradient descent from meta-learned parameters, but as noted by [10], the computation and memory demands escalate dramatically as the number of features $p$ increases, presenting significant challenges.

Recent studies [11, 12] have made initial attempts to successfully integrate meta-learning with precision matrix estimation. Despite achieving sample-efficiency through meta-learning, their methods exhibit several limitations: Restrictive Data Assumptions, and Ineffective Adaption to New Tasks. Firstly, they assume that the edges of a new task must be subsets of existing edges (where existing edges can be seen as optimized initial parameters in MAML), thereby constraining the discovery of novel connections in real-world applications. Besides, these works state that they use the state-of-the-art algorithm BigQuic, which speeds up the gradient calculation step with a block-coordinate descent Newton method, to solve a new task. The algorithm uses between $O(p)$ and $O(p^3)$ time and $O(p^2)$ and $O(p)$ memory per iteration. When the number of samples is extremely small, it takes a large number of iterations to converge to an accurate solution, leading to poor generalization performance.

Herein, we propose a novel approach for Fast and Sample-efficient Meta Precision Matrix Learning, FasMe. Our approach first extracts meta-knowledge as the shared pattern across auxiliary tasks through a multi-task learning diagram. We then adopt a maximum determinant matrix completion algorithm with meta-knowledge constraints to achieve fast adaptation in high-dimensional settings. In detail, our contributions can be summarized as follows:

- A novel meta-learning-based model: We propose a multi-task precision matrix estimator that extracts the meta-knowledge from the auxiliary tasks. We then utilize the maximum determinant matrix completion algorithm to learn a new precision matrix in the meta-testing.
- Efficient adaptation: We design a recursive closed-form algorithm to speed up the adaptation in the novel task. Our model converges to an $\epsilon$-accurate solution in $O(p \log \epsilon^{-1})$ time and $O(p)$ memory. It significantly outperforms the state-of-the-art ones.
- Theoretical guarantee: We also theoretically prove that for $p$-dimensional multivariate sub-Gaussian random vectors and $K$ auxiliary tasks with meta-knowledge $\theta$, the sample complexity of our method is $O(\log p/K)$ per auxiliary task for meta-knowledge and $O(\log |\mathcal{G}|)$ for the new task. It relaxes sample size requirements with the application of meta-learning, equipping FasMe with the ability to handle small sample settings.
- Experimental evaluation: On multiple synthetic datasets, FasMe outperforms the other baselines. Specifically, FasMe is at least ten times faster than the four baselines while promoting prediction accuracy. In the real-world experiment, FasMe obtains the minimum log-determinant Bregman divergence and performs better than the other baselines.

## 2 Background

### 2.1 Meta-Learning

The last years have witnessed a tremendous interest in methods for meta-learning, especially in a wide range of data-limited applications including medical image analysis [13, 14], language modeling [15, 16], and object detection [17, 18, 19]. Meta-learning is a machine learning technique that allows a model to learn from prior knowledge, or meta-knowledge, to improve its performance on new tasks and speed up the learning process [20, 21]. The process typically involves two stages: 1) Meta-training: In this stage, the model is trained on a set of related tasks, also known as auxiliary tasks [8]. The goal of this step is to extract meta-knowledge from these tasks, which can be used to improve the performance of the model on new tasks. This step helps the model to learn how to quickly and effectively adapt to new tasks by identifying the shared patterns or features across the auxiliary tasks [22]. 2) Meta-testing: In this stage, the model is tested on a new task, also known as the target task [23]. The goal of this step is to apply the meta-knowledge learned during the meta-training step to improve the model's performance on the target task. Meta-knowledge is used to quickly and effectively adapt the model to the new task, without the need for additional training data. Overall, meta-learning allows a model to better generalize to new tasks by leveraging prior knowledge, thereby making it more sample-efficient and fast.

### 2.2 Precision Matrix Estimation

Precision matrix estimation is the problem of finding the inverse of the covariance matrix of a multivariate random variable $\Omega = \Sigma^{-1}$. It is particularly useful in scenarios where the number of samples is equal to or smaller than the number of variables since the covariance matrix is not invertible. The graphical lasso, namely GLasso [5], is a typical penalized maximum likelihood estimator for precision matrix $\Omega$ inference under the Gaussian assumption on the dataset $\mathbf{X} \in \mathbb{R}^{n \times p}$. The popular estimator can be written as:

$$\hat{\Omega} = \arg\min_{\Omega \succ 0} -\log\det(\Omega) + <\Omega, \Sigma> + \lambda\|\Omega\|_1, \tag{1}$$

where $\lambda \geq 0$, $\Sigma = \frac{1}{n}\sum_{i=1}^{n}(\mathbf{X}_i - \bar{\mathbf{X}})(\mathbf{X}_i - \bar{\mathbf{X}})^\top$, $\bar{\mathbf{X}} = \frac{1}{n}\sum_{i=1}^{n}\mathbf{X}_i$, and $\Omega \succ 0$ denotes that $\Omega$ is positive definite and symmetric. $\|\Omega\|_1$ is the $\ell_1$-norm of matrix $\Omega$. However, [24] proved that this estimator is sensible even for non-Gaussian $\mathbf{X}$, since it corresponds to minimizing an $\ell_1$-regularized log-determinant Bregman divergence. To solve the problem Eq. (1), we take the derivative of it $\Omega^{-1} - \Sigma = \lambda\partial\|\Omega\|_1/\partial\Omega$. Inspired by this derivative, [25] proposed the constrained $\ell_1$ minimization method for inverse matrix estimation (CLIME) estimator. This estimator can be used to estimate the precision matrix $\Omega$ via an $\ell_1$ constrained optimization:

$$\arg\min_{\Omega \succ 0}\|\Omega\|_1, \quad \text{s.t. } \|\Sigma\Omega - I\|_\infty \leq \lambda. \tag{2}$$

CLIME can be solved column-by-column. Compared with GLasso, CLIME has presented more favorable theoretical properties and can be solved by column-wise linear programming.

# 3 Method

The section elaborates on how the proposed meta-learning framework works. We begin by providing a formal problem setting in Section 3.1, which covers the task and data assumptions we consider. Next, we formulate the two stages of our meta-learning framework in Section 3.2.

## 3.1 Problem Setting

**Notations** $\{\mathbf{X}^{(k)}\} = \{\mathbf{X}^{(1)}, \mathbf{X}^{(2)}, \dots, \mathbf{X}^{(K)}\}$ denotes $K$ datasets generated by $K$ auxiliary tasks. We use $\mathbf{X}^{(k)} \in \mathbb{R}^{n_k \times p}$ to represent the $k$-th dataset with $n_k$ samples and $p$ features. $\Omega^{(k)} \in \mathbb{R}^{p \times p}$ represents the $k$-th precision matrix corresponding to $\mathbf{X}^{(k)}$. $\{\Omega^{(k)}\} = \{\Omega^{(1)}, \dots, \Omega^{(K)}\}$ is the set of the precision matrices corresponding to the auxiliary data $\{\mathbf{X}^{(k)}\}$. $\Omega_{\text{new}}$ denotes the precision matrix of a new task $\mathbf{X}_{\text{new}} \in \mathbb{R}^{n \times p}$ to be estimated. We use $\theta$ to represent the meta-knowledge (i.e., the common substructure of $\{\Omega^{(k)}\}$) and $\theta_r^{(k)}$ to represent the rest of $\Omega^{(k)}$ respectively.

**Assumptions** We consider a more general class of distributions than most previous studies, i.e., sub-Gaussian distributions, which cover Gaussian variables, bounded random variables, and so on. This relaxed data assumption significantly improves the flexibility and robustness of our method. We assume that the auxiliary tasks share some meta-knowledge $\theta$ with the target task $\mathbf{X}_{\text{new}}$. Specifically, we assume that $\text{supp}(\theta) \subseteq \text{supp}(\Omega_{\text{new}})$. Here the meta-knowledge $\theta$ takes the form of a common substructure among the tasks. The assumption has been widely adopted [26, 27, 28] and proved to be feasible and applicable in the biological and genetic domains [29].

The paper aims to estimate the target precision matrix $\Omega_{\text{new}}$ of sub-Gaussian distribution with sparse structure and relatively limited samples given sufficient samples from auxiliary tasks $\{\mathbf{X}^{(k)}\}$. The auxiliary datasets follow a family of multivariate sub-Gaussian distributions (see Definition 2 in Appendix F). To address the problem, we propose a novel meta-learning framework that leverages the meta-knowledge learned from auxiliary tasks to efficiently estimate $\Omega_{\text{new}}$.

## 3.2 A Meta-learning Framework for Small Sample Precision Matrix Estimator

This part gives more details regarding the meta-learning framework we established. Figure 2 illustrates the pipeline of the framework in an intuitive way.

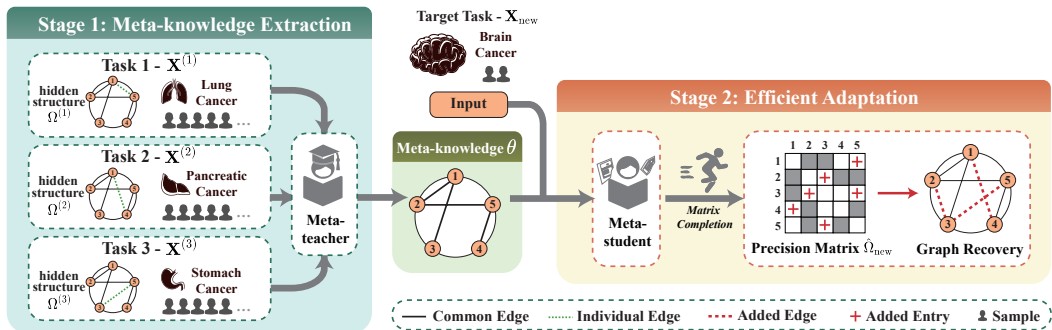

Figure 2: **The pipeline of the established framework.** The entire pipeline can be divided into two stages: meta-knowledge extraction (meta-training) and efficient adaptation (meta-testing). In Stage 1, our meta-teacher extracts the meta-knowledge, namely the shared structure, from the related auxiliary tasks with sufficient samples. In Stage 2, we obtain a prior sparsity pattern from the meta-knowledge and target dataset with a few samples. Then our meta-student aims to recover the edge structure by solving a matrix completion problem rapidly.

### 3.2.1 Meta-teacher

We extract the common substructure across the tasks as the meta-knowledge. The goal of pre-training is to extract meta-knowledge $\theta$ from auxiliary tasks $\{\mathbf{X}^{(k)}\}$.

To obtain the meta-knowledge, every precision matrix is modeled as

$$\Omega^{(k)} = \theta + \theta_r^{(k)} \tag{3}$$

where $\theta \in \mathbb{R}^{p \times p}$ is the common substructure among all graphs and $\theta_r^{(k)} \in \mathbb{R}^{p \times p}$ represents the rest for $k$-th graph.

We adopt $\ell_1$-penalization for $\theta$ and $\theta_r^{(k)}$ since they are expected to be sparse for better interpretability. Therefore, we write Eq. (3) into $\|\theta\|_1 + \rho\|\theta_r^{(k)}\|_1$, where $\rho > 0$. The value of the hyper-parameter $\rho$ depends on the properties of the graphs. This hyper-parameter controls the difference of sparsity level between $\theta$ and $\theta_r^{(k)}$. Concretely, with a smaller $\rho$, the shared part gets denser and the rest gets sparser.

Then we apply the formulation to Eq. (2), thus obtaining the single-task recovering method:

$$(\hat{\theta}, \hat{\theta}_r^{(k)}) = \underset{\theta, \theta_r}{\arg\min} \quad \|\theta\|_1 + \rho\|\theta_r^{(k)}\|_1, \qquad \text{s.t.} \quad \|\theta\Sigma^{(k)} - (I_p - \theta_r^{(k)}\Sigma^{(k)})\|_\infty \leq \lambda. \tag{4}$$

Here $\Sigma^{(k)}$ represents the covariance matrix corresponding to $\mathbf{X}^{(k)}$. This single-task method, however, ignores the inner relationship between the auxiliary tasks. To this end, we sum up all the single-task estimators and devise the optimization problem in the following form:

$$\left(\hat{\theta}, \{\hat{\theta}_r^{(k)}\}\right) = \underset{\theta, \{\theta_r^{(k)}\}}{\arg\min} K\|\theta\|_1 + \rho\sum_{k=1}^{K} \|\theta_r^{(k)}\|_1, \qquad \text{s.t.} \|\theta\Sigma^{(k)} - (I_p - \theta_r^{(k)}\Sigma^{(k)})\|_\infty \leq \lambda, \tag{5}$$

where $k = 1, 2, \ldots, K$.

### 3.2.2 Meta-student

To speed up the estimation of $\hat{\Omega}_{\text{new}}$, we propose our efficient estimator, which is favorable even with a few samples of the new task available. Additionally, we obtain a sparsity pattern from the meta-knowledge $\theta$ and target dataset with a few samples.

Different from most $\ell_1$-penalized methods, our estimator aims to estimate the precision matrix by solving a Maximum Determinant Matrix Completion (MDMC) problem. MDMC is an effective technique that aims to pick the unique maximum determinant completion (when it exists) for a partially observed matrix from all the possible matrices [30].

Consider a maximum determinant matrix completion problem for a covariance matrix with some partially observed entries:

$$\hat{\Sigma} = \underset{\Sigma \succeq 0}{\arg\max} \ \log\det\Sigma \qquad \text{s.t.} \ \Sigma_{ij} = M_{ij}, \ \forall (i,j) \in \mathcal{G}, \tag{6}$$

where $M$ represents the partially observed matrix, and the set $\mathcal{G}$ contains all the observed entries. Thus, the Lagrangian dual of this problem can be derived as:

$$\hat{\Omega} = \underset{\Omega \succeq 0}{\arg\min} \ -\log\det\Omega + \langle\Omega, M\rangle + p \qquad \text{s.t.} \ \Omega \in \mathbb{S}_{\mathcal{G}}^p, \tag{7}$$

with first-order optimality condition $|\text{sign}(M)| \odot \hat{\Omega}^{-1} = M$, where the operator $\text{sign}(\cdot)$ computes the sign of the matrix and $\odot$ denotes the Hadamard product. Herein, the set $\mathbb{S}_{\mathcal{G}}^p$ refers to the $p \times p$ symmetric matrices with sparsity pattern $\mathcal{G}$. Strong duality indicates a straightforward relation back to the primal $\hat{\Sigma} = \hat{\Omega}^{-1}$. Note that while $\hat{\Sigma}$ is (in general) a dense matrix, $\hat{\Omega}$ is always sparse. Instead of solving the primal problem for a dense matrix, we choose to solve the dual problem for a sparse matrix, which also satisfies the optimality condition.

To solve the precision matrix $\hat{\Omega}_{\text{new}}$, the first step is to find a reliable sparsity pattern. Previous works [31] obtain the pattern by thresholding the covariance matrix. However, [31] also proved the invalidity of such a method in the case of a relatively small sample size, as the covariance matrix

cannot be accurately estimated. To meet the challenge, our estimator combines the support set of the thresholded covariance matrix and meta-knowledge to obtain the sparsity pattern $\mathcal{G}$.

$$\mathcal{G} = \operatorname{supp}(\theta) \cup \operatorname{supp}(S_\eta(\Sigma_{\text{new}})). \tag{8}$$

Here, $S_\eta$ denotes the element-wise soft-thresholding operator with parameter $\eta$. Note that the problem (7) has a recursive closed-form solution whenever the graph is chordal, the embedding operation is conducted to satisfy the chordal property. To be specific, the chordal embedding of $\mathcal{G}$ is represented as $\tilde{\mathcal{G}}$ and (7) is rewritten into the following formulation,:

$$\hat{\Omega}_{\text{new}} = \underset{\Omega \succ 0}{\arg\min} - \log \det \Omega + \langle \Omega, \Pi_\mathcal{G}(\Sigma_{\text{new}}) \rangle, \qquad \text{s.t. } \Omega_{ij} = 0, \ (i,j) \in \tilde{\mathcal{G}} \backslash \mathcal{G}. \tag{9}$$

where $\Pi_\mathcal{G}$ is the projection operator from $\mathbb{S}^p$ onto $\mathbb{S}^p_E$, i.e., by setting $\Pi_\mathcal{G}(A_{ij}) = 0$ if $(i,j) \notin E$.

### 3.3 Optimization Algorithms

The learning steps of meta-knowledge extraction (5) is solved efficiently through a formulation of multiple independent sub-problems of linear programming, which can be accelerated in a parallel form. With the assistance of meta-knowledge, we solve the optimization problem (9) using Newton Conjugate Gradient method. The key idea is to construct an inner conjugate gradients loop as a solution to the Newton subproblem of an outer Newton's method. The detailed description, complexity analysis, and pseudo-code (Algorithm 1) of the full algorithm are shown in Appendix C.

## 4 Theoretical Analysis

To present the analysis more concisely, we first define $\Sigma_{\text{tot}} := \operatorname{diag}(\Sigma^{(1)}, \ldots, \Sigma^{(K)}) = (\sigma_{ij})_{Kp \times Kp}$, $\Theta := \operatorname{diag}(\theta, \ldots, \theta)$, $\Theta_r := \operatorname{diag}(\theta_r^{(1)}, \ldots, \theta_r^{(K)})$, $\Omega_{\text{tot}} := \operatorname{diag}(\Omega^{(1)}, \ldots, \Omega^{(K)}) = \Theta + \Theta_r$, $\mathbf{X}_{\text{tot}} := \operatorname{diag}(\mathbf{X}^{(1)}, \ldots, \mathbf{X}^{(K)})$. We assume that the precision matrix $\Omega$ belongs to the uniformity class of matrices,

$$\mathcal{U} := \mathcal{U}(q, s(Kp)) = \left\{ \Omega : \Omega \succ 0, \|\Omega\|_1 \leq \nu, \|\Omega\|_\infty \leq \phi, \max_{1 \leq i \leq Kp} \sum_{j=1}^{Kp} |\omega_{ij}|^q \leq s(Kp) \right\}. \tag{10}$$

Here $q, \nu, \phi$ are some constants, $0 \leq q < 1$ and $\Omega := (w_{ij})_{Kp \times Kp}$. $s(Kp)$ represents the sparsity level of $\Omega$ in the uniformity class. Note that the sparsity level is related to $p$ without an analytic form of relationships between them.

Then some important conditions and definitions are stated as the following.

**Exponential Tail Condition**  Suppose there exists a constant $0 \leq \gamma \leq \frac{1}{4}$, so that $\frac{\log p}{n_k} \leq \gamma$ and

$$\mathbb{E}[\exp(t(X_i - \mu_i)^2)] \leq C \leq \infty, \forall |t| \leq \gamma, \forall i \in \{1, \ldots, p\}, \tag{11}$$

where $C$ is a constant.

**Irrepresentable Condition**  [4] There exists some $\alpha \in (0, 1]$ such that

$$\||\Gamma^*_{\mathcal{G}^c\mathcal{G}}(\Gamma^*_{\mathcal{G}\mathcal{G}})^{-1}\||_1 \leq 1 - \alpha, \tag{12}$$

where $\Gamma^* := \Omega^{*-1} \otimes \Omega^{*-1}$ ($\otimes$ represents the Kronecker matrix product) and $\mathcal{G}^c = \{1, \ldots, p\}^2 \backslash \mathcal{G}$.

We define $\kappa_{\Gamma^*} = \||(\Gamma^*_{\mathcal{G}\mathcal{G}})^{-1}\||_\infty$. Note that $\Gamma^*_{\mathcal{G}\mathcal{G}}$ is a $(s+p) \times (s+p)$ matrix indexed by vertex pairs, where $s = |\mathcal{G}|$. We define $\omega_{\min}^{\text{new}} = \min_{(i,j) \in \operatorname{supp}(\Omega^*_{\text{new}})} |\Omega^*_{\text{new},ij}|$ and $\omega_{\min}^{\text{tot}} = \min_{(i,j) \in \operatorname{supp}(\Omega^*_{\text{tot}})} |\Omega^*_{\text{tot},ij}|$.

**Definition 1.** *[32] Given a matrix $M \in \mathbb{S}^p$, define $\mathcal{G}_M = \{(i,j) : M_{ij} \neq 0\}$ as its sparsity pattern. Then $M$ is called **inverse-consistent** if there exists a matrix $N \in \mathbb{S}^p$ such that*

$$M + N \succeq 0, \quad \forall (i,j) \in \mathcal{G}_M \quad N_{ij} = 0, \quad (M+N)^{-1} \in \mathbb{S}^p_{\mathcal{G}_M} \tag{13}$$

*The matrix $N$ is called an inverse-consistent complement of $M$ and is denoted by $M^{(c)}$. Furthermore, $M$ is called **sign-consistent** if for every $(i,j) \in \mathcal{G}_M$, the $(i,j)$-th elements of $M$ and $(M + M^{(c)})^{-1}$ have opposite signs.*

Then we define the $\beta(\mathcal{G}, \alpha)$ function with respect to the sparsity pattern $\mathcal{G}$ and scalar $\alpha > 0$

$$\beta(\mathcal{G}, \alpha) = \max_{M \succ 0} \quad \|M^{(c)}\|_{\max} \quad \text{s.t. } M \in \mathbb{S}_{\mathcal{G}}^n \text{ and } \|M\|_{\max} \leq \alpha, M_{i,i} = 1, M \text{ is inverse-consistent.}$$

Here $i = 1, 2, \ldots, n$, $\|\cdot\|_{\max}$ denotes the $\ell_{\max}$-norm, e.g., $\|A\|_{\max} = \max_{i \neq j} |A_{ij}|$.

The Appendix F provides all the detailed proofs of the lemmas, theorems, and corollaries.

## 4.1 Main Theorems

In this section, we mainly study the theoretical properties of the meta-knowledge $\hat{\theta}$ in (5) and the precision matrix $\hat{\Omega}_{\text{new}}$ in (7). The first theorem specifies a probability lower bound of recovering the true meta-knowledge by our estimator in (5) for multiple random multivariate sub-Gaussian distributions.

**Theorem 1.** *Suppose that $\Omega_{\text{tot}}^* \in \mathcal{U}$ and $N = \sum_{k=1}^K n_k$. When the variables are sub-Gaussian, the tail condition holds. Let $\lambda = C_0 \nu \sqrt{\frac{\log(Kp)}{N}}$, where the constant $C_0 = 2\gamma^{-2}(2 + \tau_0 + \gamma^{-1}e^2C^2)^2$ and constant $\tau_0 > 0$. If $\omega_{\min}^{\text{tot}} > 4\tau_n$, then we have*

1. $\|\hat{\theta} - \theta^*\|_\infty \leq 8C_0\nu^2 \sqrt{\frac{\log(Kp)}{N}} + 2\phi$;

2. $\text{supp}(\tilde{\theta}) = \text{supp}(\theta^*)$;

*with a probability greater than $1 - 4p^{-\tau_0}$. Here $\tilde{\theta}$ is a threshold estimator with $\tilde{\theta}_{ij} = \hat{\theta}_{ij}I(|\hat{\theta}_{ij}| \geq \tau_n)$, where $\tau_n \geq 2\nu\lambda$ is a tuning parameter.*

According to Theorem 1, if $n_1 = n_2 = \cdots = n_k$, a sample complexity $O(\frac{\log K + \log p}{K})$ per task is sufficient for the recovery of the meta-knowledge. In most cases, $O(\frac{\log K + \log p}{K}) \to O(\frac{\log p}{K})$ since $p >> K$.

The following theorem specifies a probability lower bound of recovering a correct precision matrix $\hat{\Omega}_{\text{new}}$ by our estimator in (7) for a multivariate sub-Gaussian distribution.

**Theorem 2.** *Suppose we have recovered the true sparsity pattern $\mathcal{G}$ of a family of $p$-dimensional random multivariate sub-Gaussian distribution. Then for a new task of multivariate sub-Gaussian distribution with the precision matrix $\Omega_{\text{new}}^*$ such that $\text{supp}(\theta) \subseteq \text{supp}(\Omega_{\text{new}}^*)$ and satisfying irrepresentable condition, consider the estimator $\hat{\Omega}_{\text{new}}$ with $\lambda = C_3\nu\sqrt{\frac{\log|\mathcal{G}|}{n}}$. Then we have*

$$\|\hat{\Omega}_{\text{new}} - \Omega_{\text{new}}^*\|_\infty \leq 2C_3\nu^2\kappa_{\Gamma^*}\sqrt{\frac{\log|\mathcal{G}|}{n}}, \tag{14}$$

*with a probability $1 - p^{-\tau_1}$, where $\tau_1 > 0$.*

This theorem shows that $n \in O(\log|\mathcal{G}|)$ is sufficient for estimating a correct precision matrix of the new task with our estimation. Then, efforts should be made to prove that the max-det matrix completion method guarantees the consistency of the support set of $\hat{\Omega}_{\text{new}}$ with $\mathcal{G}$.

**Theorem 3.** *$\mathcal{G}$ coincides with the sparsity pattern of the optimal solution $\Omega_{\text{new}}^*$ if the normalized matrix $\tilde{\Sigma}_{\text{new}} = D^{-1/2}\Pi_{\mathcal{G}}(\Sigma_{\text{new}})D^{-1/2}$ where $D = \text{diag}(\Pi_{\mathcal{G}}(\Sigma_{\text{new}}))$ satisfies the following conditions:*

1. *$\tilde{\Sigma}_{\text{new}}$ is positive definite and sign-consistent,*

2. *We have*

$$\beta\left(\mathcal{G}, \|\tilde{\Sigma}_{\text{new}}\|_{\max}\right) \leq \min_{(k,l) \notin \mathcal{G}} \frac{\eta - |(\Pi_{\mathcal{G}}(\Sigma_{\text{new}}))_{kl}|}{\sqrt{(\Pi_{\mathcal{G}}(\Sigma_{\text{new}}))_{kk} \cdot (\Pi_{\mathcal{G}}(\Sigma_{\text{new}}))_{ll}}}. \tag{15}$$

We set the diagonal elements of $\tilde{\Sigma}_{\text{new}}$ to 1 and its off-diagonal elements between -1 and 1. The projection operation $\Pi_{\mathcal{G}}(\cdot)$ leads to many zero elements in $\tilde{\Sigma}_{\text{new}}$, resulting in $\tilde{\Sigma}_{\text{new}}$ being positive

definite or even diagonally dominant in most cases. As shown by [32], when $\mathcal{G}$ induces an acyclic structure, condition 2 is automatically satisfied by condition 1. More generally, [33] shows that $\tilde{\Sigma}_{\text{new}}$ is sign-consistent if $(\tilde{\Sigma}_{\text{new}} + \tilde{\Sigma}_{\text{new}}^{(c)})^{-1}$ is close to its first order Taylor expansion. This assumption holds in practice due to the fact that the magnitude of the off-diagonal elements of $\tilde{\Sigma}_{\text{new}} + \tilde{\Sigma}_{\text{new}}^{(c)}$ is small. Moreover, [32] shows that the left side of (50) is upper bounded by $a \cdot \|\tilde{\Sigma}_{\text{new}}\|_{\max}^2$ for some $a > 0$. That implies that when $\|\tilde{\Sigma}_{\text{new}}\|_{\max}$ is small, or equivalently $\eta$ is large, condition 3 is automatically satisfied. If the conditions in Theorem 3 are satisfied, the support set of $\hat{\Omega}_{\text{new}}$ is consistent with $\mathcal{G}$ according to (7).

## 5 Related Work

There have been several methods proposed to estimate precision matrices, among which, three state-of-the-art methods, QUIC [10], Neighborhood Selection [34], Meta-IE [11] and gRankLasso [35] are chosen as baselines in the experiments (See reasons in Appendix E). These approaches all consider high-dimensional settings which are similar to our work. **QUIC:** This approach utilizes a quadratic approximation of the negative log-likelihood function to estimate sparse inverse covariance matrices to accelerate computation speed, which reduces the computational cost from $O(p^2)$ to $O(p)$ at each iteration. **Neighborhood Selection:** This method is a type of regularized maximum likelihood estimation method for estimating precision matrices. Given some i.i.d. observations, it estimates the conditional independence restrictions separately for each node in the graph. The complexity of the neighborhood selection for one node with the Lasso is $O(np \min\{n, p\})$ using LARS algorithm [36]. **Meta-IE (BigQuic):** [11] first proposes a meta-learning-based method that is composed of two steps. First, Meta-IE takes the support union of all the precision matrices of the auxiliary tasks as prior knowledge. Then the method states that the testing step can be solved by an interior point method [37] in polynomial time $O(p^{3.5} \log \epsilon^{-1})$ or BigQuic [38] with a better time complexity around $O(p^2 \log \epsilon^{-1})$ (See analysis in Appendix). **gRankLasso:** gRankLasso [35] is a method for sparse precision matrix estimation that realizes automatic parameter tuning, making it completely tuning-free. It achieves robustness and accuracy by integrating a group penalty with Lasso regularization, effectively estimating sparse structures in high-dimensional data.

## 6 Experiment

### 6.1 Synthetic Experiment

**Baselines:** We compare our method with the following baselines mentioned in Section 5: 1) The QUIC baseline, 2) the Neighborhood Selection (NS) baseline, 3) Meta-IE (BigQuic) baseline that is accelerated with 32 threads and 4) gRankLasso baseline.

**Metric:** We use the Frobenuis-norm $\|\hat{\Omega}_{\text{new}} - \Omega_{\text{new}}^*\|_F$ and Mathews correlation coefficient (MCC) as metrics to evaluate the performance. MCC is widely used in machine learning as a measure of binary classifiers, defined as $\text{MCC} = (\text{TP} \times \text{TN} - \text{FP} \times \text{FN})/\sqrt{(\text{TP} + \text{FP})(\text{TP} + \text{FN})(\text{TN} + \text{FP})(\text{TN} + \text{FN})}$. Here the true positive (TP), true negative (TN), false positive (FP), and false negative (FN) values indicate the number of true non-zero entries, true-zero entries, false non-zero entries, and false zero entries, respectively. It produces a high score if the classifier generates desirable estimations.

**Simulated Datasets:** Each graph model contains a new task and $K = 4$ prior learning tasks. The sample matrices follow multivariate normal distributions where the corresponding precision matrices are generated using two sparse models: (1) Random graph and (2) Tree graph. More details about the data simulation are moved into Appendix B.

**Time Comparison** The time-complexity experiment on simulated datasets is conducted to observe the changes of all the estimators in computation time with the varying $p$ and $n$. As shown in Fig. 3 and Table 1 &2, FasMe performs better than all the baselines. Using our method, we solve a sparse inverse covariance estimation problem containing 2500 variables, with time cost around 1 second. By comparing all the subfigures in Fig. 3, we find that all the methods except the neighborhood selection approach are insensitive to the change of $n$. This is because the neighborhood selection deals with the sample matrix $\mathbf{X}_{\text{new}} \in \mathbb{R}^{n \times p}$ directly instead of the covariance matrix $\Sigma_{\text{new}} \in \mathbb{R}^{p \times p}$. However, it remains computationally costly compared to FasMe when the sample size is much

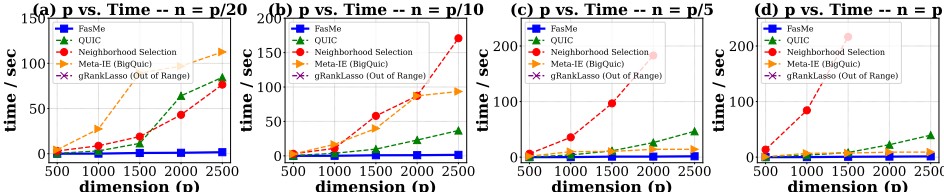

Figure 3: Time cost of FasMe vs. baselines on simulated datasets with the feature dimension $p$ varying in $\{500, 1000, 1500, 2000, 2500\}$. Subfigure (a)(b)(c)(d) records the time required for each method to be implemented on a series of datasets with different sample sizes $n = p/20, p/10, p/5, p$, respectively. Note that the missing points of the baselines mean the time cost is out of range.

smaller than the number of features. Interestingly, despite Meta-IE (BigQuic) method can speed up by parallelizing multiple threads, it is less time-efficient because of the time assumption of the I/O reads and writes. Moreover, BigQuic cannot converge within the maximum iterations ($\mathrm{maxiter} = 100$), especially when only a few samples are provided ($n = p/20, p/10$). Regarding gRankLasso, its tuning mechanism requires a substantial amount of time, leading to significant time costs in high-dimensional scenarios. In contrast, our method converges significantly faster within 20 iterations, regardless of small sample setting.

Table 1: Comparison of estimation error (in terms of MCC and F-norm= $\|\hat{\mathbf{\Omega}}_{\mathrm{new}} - \mathbf{\Omega}^*_{\mathrm{new}}\|_F$) and running time (seconds) on synthetic dataset. $\Omega^*$ is generated from a Random graph. $*$ means that the time exceeds 30 minutes.

| Methods | $n = 50, p = 1000$ | | | $n = 100, p = 1000$ | | | $n = 100, p = 2000$ | | | $n = 200, p = 2000$ | | |
|---|---|---|---|---|---|---|---|---|---|---|---|---|
| | MCC($\uparrow$) | F-norm($\downarrow$) | Time($\downarrow$) | MCC($\uparrow$) | F-norm($\downarrow$) | Time($\downarrow$) | MCC($\uparrow$) | F-norm($\downarrow$) | Time($\downarrow$) | MCC($\uparrow$) | F-norm($\downarrow$) | Time($\downarrow$) |
| QUIC | 0.043 | 23.02 | 3.028 | 0.065 | 22.46 | 3.676 | 0.039 | 33.87 | 64.062 | 0.065 | 31.65 | 22.925 |
| NS | 0.287 | 66.08 | 8.724 | 0.436 | 54.21 | 10.829 | 0.388 | 88.98 | 43.003 | 0.624 | 67.02 | 87.193 |
| Meta-IE (BigQuic) | 0.403 | 29.10 | 27.370 | 0.460 | 22.42 | 16.655 | 0.281 | 41.92 | 96.850 | 0.310 | 32.10 | 87.140 |
| gRankLasso | 0.008 | 198.56 | $*$ | 0.012 | 176.99 | $*$ | 0.005 | 218.12 | $*$ | 0.008 | 212.43 | $*$ |
| Ours | 0.659 | 22.63 | 0.061 | 0.708 | 19.37 | 0.063 | 0.661 | 26.01 | 0.994 | 0.716 | 23.13 | 0.965 |

Table 2: Comparison of estimation error (in terms of MCC and F-norm= $\|\hat{\mathbf{\Omega}}_{\mathrm{new}} - \mathbf{\Omega}^*_{\mathrm{new}}\|_F$) and running time (seconds) on synthetic dataset. $\Omega^*$ is generated from Tree graph.

| Methods | $n = 50, p = 1000$ | | | $n = 100, p = 1000$ | | | $n = 100, p = 2000$ | | | $n = 200, p = 2000$ | | |
|---|---|---|---|---|---|---|---|---|---|---|---|---|
| | MCC($\uparrow$) | F-norm($\downarrow$) | Time($\downarrow$) | MCC($\uparrow$) | F-norm($\downarrow$) | Time($\downarrow$) | MCC($\uparrow$) | F-norm($\downarrow$) | Time($\downarrow$) | MCC($\uparrow$) | F-norm($\downarrow$) | Time($\downarrow$) |
| QUIC | 0.038 | 22.98 | 3.212 | 0.071 | 20.69 | 3.541 | 0.036 | 35.07 | 58.972 | 0.070 | 31.48 | 31.625 |
| NS | 0.306 | 66.02 | 8.701 | 0.485 | 56.21 | 11.147 | 0.396 | 87.87 | 50.601 | 0.655 | 66.08 | 89.237 |
| Meta-IE (BigQuic) | 0.451 | 28.64 | 26.321 | 0.482 | 20.88 | 17.632 | 0.305 | 39.97 | 99.789 | 0.340 | 31.52 | 85.176 |
| gRankLasso | 0.007 | 201.89 | $*$ | 0.013 | 185.60 | $*$ | 0.006 | 223.45 | $*$ | 0.008 | 205.77 | $*$ |
| Ours | 0.668 | 22.05 | 0.058 | 0.712 | 15.98 | 0.064 | 0.681 | 24.24 | 1.002 | 0.745 | 24.02 | 0.996 |

**Accuracy Comparison** We conduct several experiments on the simulated datasets to compare the prediction power of all methods with four different small sample settings $n < p/10$. Table 1&2 indicates that FasMe obtains lower Frobenius-norm and higher MCC under all the conditions. It suggests that FasMe outperforms the other baselines in high-dimensional settings. The comparison of the two tables shows that FasMe can effectively recover the special structure from the data. In addition, the MCC of Meta-IE becomes small as the feature dimension increases. This is because the predicted edges must be a subset of the support union. As $p$ increases, the estimation of the support union is not relatively accurate, thus significantly reducing the prediction accuracy of Meta-IE. The poor performance of gRankLasso can be attributed to the inefficacy of its automatic tuning mechanism in small sample settings, which degrades the model performance. We draw one subnetwork for the experimental result of every method and compare them with the ground truth in Fig. 4 (a)(b). They show that our method bears the highest similarity with the ground truth. The results are consistent with the MCC value of every method shown in Table 1 &2.

## 6.2 Real-world Experiment: Application to Gene and fMRI Data

We further evaluate our method for estimating precision matrices on two real-world datasets: ChIP-Seq dataset (ENCODE project [39]) and fMRI dataset (OpenfMRI project [40], accession number: ds000002). Regarding the ChIP-Seq dataset, we aim to exploit the gene network of transcription

factors (TFs). We randomly selected 300 gene features. 27 samples of H1-hESC (embryonic stem cells) and GM12878 (Blymphocyte) are chosen as two auxiliary tasks for meta-knowledge and 20 samples of K562 (leukemia) are chosen as a novel task. Regarding the fMRI dataset, we aim to exploit brain connectivity among different brain regions. We randomly selected 200 regions of interest (ROI) as features. 34 samples of PCL and DCL datasets are chosen as the auxiliary tasks for meta-knowledge and 20 samples of Mixed-Event dataset are chosen as a novel task. Additional experimental details including configuration choices and extended experiments in the economic domain are provided in Appendix B.

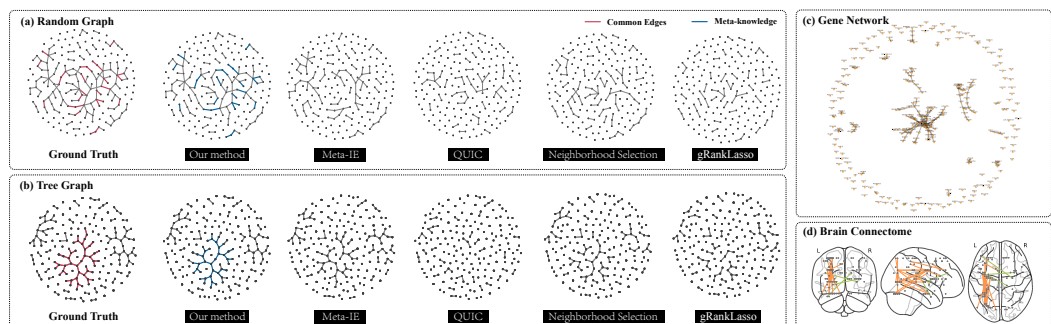

Figure 4: Subfigure (a)(b) display graph recovery results on two synthetic datasets for different methods compared with the ground truth. Subfigure (c) shows the gene network predicted by our proposed method for 300 genes on ChIP-Seq dataset. Subfigure (d) shows the brain connectome recovered by our proposed method for 200 regions on fMRI dataset. Positive and negative correlations are represented by orange and green edges, respectively.

In Subfigure (c)(d) of Fig. 4 , we visualize the experimental results on the two real-world datasets respectively. The predicted gene network exhibits scale-free and clustering behaviors, which is consistent with its biological properties. In another experiment, the majority of brain connections are found in the left ROIs. Besides, the left brain has positive connections, while the cross-hemisphere ones are negative. In this case, the subjects are asked to solve a classification problem, which mainly relies on their left brain's analysis function. The results align with our expectations.

## 7    Conclusion

In this work, we introduced FasMe, a Fast and Sample-efficient Meta Estimator designed to address the challenges of precision matrix learning in small sample settings. Under a novel meta-learning-based framework, our approach first leverages a multi-task precision matrix estimator to extract shared meta-knowledge from auxiliary tasks, enabling efficient adaptation to new tasks with minimal data. FasMe then incorporates a maximum determinant matrix completion strategy to enhance precision matrix estimation, ensuring both computational efficiency and theoretical robustness. Our experiments on synthetic and real-world datasets demonstrate that FasMe significantly outperforms state-of-the-art baselines in terms of accuracy and speed. These results highlight the potential of FasMe to address real-world challenges in high-dimensional settings, such as genetics and neuroscience, where data is scarce. Moving forward, we plan to extend our work by relaxing the sub-Gaussian data assumption and further reducing the sample size requirements for training meta-knowledge. This will extend the framework to other domains and improve scalability for even larger datasets.

## Acknowledgements

This work was supported by National Natural Science Foundation of China (Grant Numbers 61906040, 61972085, 62276063, 62272101, 6509009710), the National Key Research and Development Program of China (Grant Number 2022YFF0712400), the Natural Science Foundation of Jiangsu Province (Grant Number BK20221457, BK20230083), the Fundamental Research Funds for the Central Universities (Grant Number 2242021R41177), and SIP Support-startup funding (Grant Number MSRI8001004).

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

# A  Impact Statements

This paper proposes a novel approach for fast and sample-efficient meta precision matrix learning, which can leverage prior knowledge from related tasks to improve the estimation of conditional dependency relationships between random variables. This approach has potential applications in various domains such as genetics, neuroscience, and social networks, where sparse and high-dimensional data are common. The ethical aspects and future societal consequences of this work depend on the specific contexts and purposes of applying this approach, which are beyond the scope of this paper. However, we encourage researchers and practitioners to consider the possible benefits and risks of using this approach, such as enhancing disease diagnosis or enabling authoritarian surveillance, and to adopt appropriate measures to ensure its responsible and beneficial use.

# B  More Experimental Details

All experiments are performed on a machine with an Intel Core i9-10910 ten-core 3.6 GHz CPU and 64 GB RAM.

## B.1  Simulation Experiment

For each prior task $k = 1, 2, \ldots, K$, we generate $n_k = p/2$ independently and identically distributed observations from a multivariate normal distribution with mean $\mathbf{0}$ and precision matrix $\Omega^{(k)}$. For the new task, we generate $n = p/10$ i.i.d. samples from a Gaussian distribution with mean $\mathbf{0}$ and precision matrix $\Omega_{\text{new}} = \Omega^{(k+1)}$. This model assumes that the graph is composed of two parts, namely the common substructure $\theta$ and the rest $\theta_r^{(k)}$. The entire precision matrix is parameterized as $\Omega^{(k)} = \theta + \theta_r^{(k)}$.

**Random Graph**   Each off-diagonal entry of $\theta$ is generated independently and equals $0.5$ with a probability $p_0 = 1/(p-1)$ and $0$ with a probability $1 - p_0$. Each off-diagonal entry of $\theta_r^{(k)}$ is generated independently and equals $0.5$ with probability $0.2kp_0$ and $0$ with probability $1 - 0.2kp_0$. Then we add every diagonal element by a fixed value large enough to guarantee the positive definiteness of the precision matrix.

**Tree Graph**   We generate $K + 1$ networks each consisting of $c$ unconnected subnetworks with a tree structure. All the networks share one common subnetwork with $p/10$ dimensions. We randomly generate the dimension of the other $c - 1$ subnetworks that range from 1 to $p/10$. Every subnetwork follows a tree structure. Most subnetworks consist of only one node because we choose a relatively large value for $c$. Each off-diagonal entry corresponding to the edge equals $0.5$. Then we add every diagonal element by a fixed value large enough to guarantee the positive definiteness of the precision matrix.

## B.2  Real-world Experiment

**ChIP-Seq dataset**   ENCODE project is a widely used database for human genome research. For our experiment, we select the ChIP-Seq data for transcription factors (TFs). The ChIP-Seq dataset has been widely utilized as the benchmark dataset in other research papers in this domain [41, 42, 43, 44]. TFs are proteins that can control how genes work. The function of TFs is to regulate turn-on and off genes in order to make sure that they are expressed in the desired cells at the right time and in the right amount throughout the life of the cell and the organism. This experiment aims to exploit the gene network of transcription factors (TFs).

There are three kinds of human cells involved in the dataset: (1) H1-hESC(embryonic stem cells: primary tissue), (2) GM12878 (B-lymphocyte: normal tissue) and (3) K562 (leukemia: cancerous tissue). The dataset provides simultaneous binding measurements of TFs to thousands of gene targets, each file contains 27 samples(TFs) and each sample has 25185 features(gene targets). We randomly selected 300 features for our experiment. The exact shape of the used data in our experiment is (27 TF samples, 300 TF features) for each kind of human cell. H1-hESC and GM12878 are chosen as two auxiliary tasks for meta- knowledge and 20 samples of K562 are chosen as a new task.

Note that the choice to utilize H1-hESC and GM12878 as auxiliary tasks, with K562 as the primary prediction task, was made with deliberate consideration of their biological significance. H1-hESC represents primary tissue, GM12878 represents normal tissue, and K562 represents cancer tissue. Given the heightened research focus on cancer cell gene networks, which are more likely to be the subject of predictive modeling and have rare cases, we opted for this configuration to use K562 as the test set.

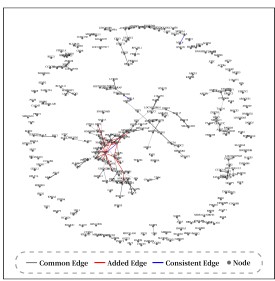

Figure 5: Gene network result of FasMe. Examples of discovered gene relationships that are consistent with other biological research: MYL6-GAA [45], HIF1A-ATP6V1G1 [46], WDR6-RBM4 [47].

**fMRI dataset** OpenfMRI is a task-based fMRI database, we chose a dataset about the task Classification learning (accession number: ds000002). This database has been widely used in other research papers in this domain [48, 49, 50, 51]. This dataset contains three different tasks, participants were instructed to perform probabilistic classification learning (PCL), deterministic classification learning (DCL), and mixed-up classification learning on Weather Prediction Task(WPT). WPT is commonly used to assess striatal or procedural learning capacities in different populations. This fMRI dataset effectively records neural responses to stimuli, delays, and negative and positive feedback components, and studying this dataset aims to address a fundamental question of whether and how the brain's memory systems interact.

In our experiment, the 4D fMRI brain image is divided into spatial regions of size $2 \times 2 \times 2$, so the original scan data in spatial shape $64 \times 64 \times 30$ is divided into 15360 regions. And the activation of each region during the task (the temporal dimension) is added up to represent the average activation situation of corresponding regions. We randomly selected 200 active regions to lower the computational burden. By training on the tasks of PCL and DCL, the common key brain regions and the interaction between the tasks are obtained, that is, a common substructure.

Note that we employed PCL (Probabilistic Classification Learning) and DCL (Deterministic Classification Learning) as auxiliary datasets due to their representation of distinct learning paradigms (See Appendix A.2). PCL captures learning under uncertainty, while DCL reflects learning from clear-cut rules. This diversity aids in generalizing the model to novel scenarios. The Mixed-Event dataset, embodying elements from both PCL and DCL, mirrors the intricate and multifaceted nature of human cognition. Consequently, the Mixed-Event dataset emerged as the optimal choice for the test set, offering a realistic and challenging environment for the model to demonstrate its predictive prowess in brain connectivity, reflective of the complexity encountered in real-world cognitive processes.

By using the prior knowledge of common substructure, we effectively introduce the key areas of the brain on similar tasks and the interaction between these key areas, so that we can learn how the brain regions interact in other tasks efficiently.

In Table 3, we record the negative log-determinant Bregman divergence of our estimator for the new task and compare it with the other baselines. The experimental results show that our method generalizes better than all other baselines for the real-world dataset since it achieves the minimum log-determinant Bregman divergence.

**US Industry Portfolios dataset** This is a well-known dataset of US industry portfolios' monthly returns from 1926 to 2023. It covers various economic sectors such as agriculture, healthcare, energy, machinery, and electronics. We take each industry as a feature ($p = 49$) and use the records of different financial metrics as the tasks. We use AVWR (Annual Voluntary Wage Reporting) dataset

Table 3: Negative log-determinant Bregman divergence of the estimated precision matrices of the new task in the real-world dataset using FasMe and other baselines. A larger value of negative log-determinant Bregman divergence indicates better performance.

| Method | Negative log-determinant Bregman divergence | |
| --- | --- | --- |
| | fMRI Data | ChIP-Seq Data |
| QUIC | -2001.26 | -716.32 |
| Neighborhood Selection | -284.35 | -316.21 |
| Meta-IE | -291.66 | -200.45 |
| gRankLasso | -2982.78 | -823.91 |
| FasMe | **-56.31** | **-64.35** |

and AEWR (Adverse Effect Wage Rate) dataset ($n_1, n_2 = 50$) for meta-training, and AFS (Annual Financial Statement) dataset for meta-testing ($n = 10$). With FasMe, we can extract meta-knowledge about industry interactions from records of different financial metrics, thus resulting in a better estimation of the hidden relationships of industries in the new task. The predicted connections, e.g., Steel (Steel Works)-Trans(Transportation) and ElcEq (Electrical Equipment)-Hardw(Computers), are consistent with reality.

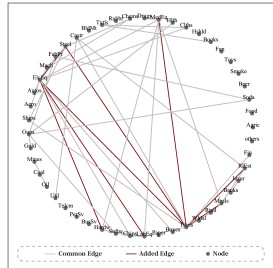

Figure 6: Visualization of the financial network predicted by FasMe.

## B.3 Time Complexity Analysis of Meta-IE (BigQuic)

BigQuic [38], an "big-data" extension of QUIC, aims to estimate the precision matrix with thousands of variables. It solves the optimization problem using a block-coordinate descent Newton method. BigQuic divides the Newton direction into several blocks and utilizes the memory cache to accelerate the updated process. The computation complexity of BigQuic is mainly determined by $O\left((p + |B|)hTT_{\text{outer}}\right)$, where $|B|$ is the number of boundary nodes, $h$ is the number of non-zero entries in the $t$-th generation estimated solution $\hat{\Omega}^{(t)}$, $T$ is the average number of Conjugate Gradient iterations and $T_{\text{outer}}$ is the number of computations within a block. Without loss of generality, $h$ is larger than $p$ in $t$-th iteration. Hence, the time complexity of BigQuic is at least higher than $O(p^2)$ time complexity. Furthermore, the performance of BigQuic depends largely on the choice of cluster scheme. A poor partition of variables leads to many "cache misses". The challenges of choosing a cluster schema make BigQuic harder to implement.

## B.4 Difference between Meta-IE and FasMe

Based on the meta-learning framework, FasMe and Meta-IE differ in each component of this framework, namely meta-teacher, meta-knowledge, and meta-student.

### B.4.1 Meta-knowledge

**Meta-IE:** Meta-IE supposes the support union $\hat{\Omega}$ as the meta-knowledge and assumes that the edges of the new task must be a subset of the edges of $\hat{\Omega}$.

**FasMe:** Different from Meta-IE, we assume that the new task shares a common structure, i.e., meta-knowledge $\hat{\theta}$, with the related auxiliary tasks.

As we stated in Section Introduction, it is impossible to make startling discoveries under this unrealistic assumption of Meta-IE. Our choice of meta-knowledge is more reasonable with biological research and existing works as evidence.

### B.4.2  Meta-teacher

**Meta-IE:** Meta-IE simply pools all the samples from $K$ auxiliary tasks to learn the meta-knowledge:

$$\hat{\Omega} = \arg\min_{\Omega \succeq 0} -\log\det\Omega + \left\langle \sum_{k=1}^{K} \frac{1}{K}\Sigma^{(k)}, \Omega \right\rangle + \lambda\|\Omega\|_1,$$

which is data-inefficient.

**FasMe:** By taking the inherent heterogeneity among the auxiliary tasks into consideration, FasMe proposes a multitask-learning-based estimator to extract the common substructure $\hat{\theta}$ across the tasks.

### B.4.3  Meta-student

**Meta-IE:** Meta-IE uses BigQUIC method to solve a GLasso problem with the knowledge constraint $\text{supp}(\Omega) \subseteq \text{supp}(\hat{\Omega})$.

**FasMe:** FasMe aims to estimate the precision matrix by solving the dual form of a MDMC problem with Newton Conjugate Gradient algorithm. We also would like to emphasize that the meta-student is not a simple application of standard MDMC. The formulation of a standard MDMC problem is:

$$\max_{X} \quad \log\det X \quad \text{s.t.} \quad X_{ij} = M_{ij}, \forall i,j \in E.$$

We improve the method from two aspects: 1) able to deal with the partially observed matrices without chordality by utilizing the chordal embedding 2) lower the time complexity by applying Newton CG algorithm

Table 4: Comparison of standard MDMC and our improved MDMC

|  | **standard MDMC** | **our improved MDMC** |
|---|---|---|
| **Targeted matrix** | partially observed matrices with chordality | partially observed matrices |
| **Time complexity** | $O(p^2 O_{\text{INV}})$ where $O_{\text{INV}}$ is the time required for matrix inversion | $O(p\log\epsilon^{-1})$ |

In summary, Meta-IE has presented a preliminary idea for applying metalearning to precision matrix estimation with much potential for improvement. Compared to Meta-IE, FasMe exhibits better theoretical properties and superior performance.

### B.5  Accuracy with Different Sparsity

Table 5: Random graph, $n = 100, p = 1000$

| Sparsity | MCC score (%) |
|---|---|
| $1/p$ | 70.8 |
| $10/p$ | 70.5 |
| $20/p$ | 71.2 |
| $30/p$ | 70.3 |
| $40/p$ | 72.1 |
| $50/p$ | 71.5 |

Table 6: Tree graph, $n = 100, p = 1000$

| Sparsity | MCC score (%) |
|----------|---------------|
| $1/p$ | 71.2 |
| $10/p$ | 70.9 |
| $20/p$ | 71.6 |
| $30/p$ | 72.1 |
| $40/p$ | 71.7 |
| $50/p$ | 72.3 |

## B.6 Hyperarameter Selection

$K$   We actually have conducted a synthetic experiment with varying $K$ and a fixed data size ($n = 100, p = 1000$). We use MCC value as a metric to measure the performance of estimating meta-knowledge. A higher score represents better performance. By varying $K = 2, 3, \ldots, 6$, Fig. 7 shows that the MCC value smoothly experiences a slight decline from $0.85$ to $0.60$.

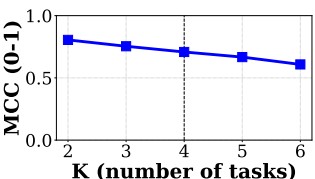

Figure 7: MCC values of our methods against varying $K$

$\rho$   This hyperparameter delineates the sparsity contrast between the shared and individual components of the model. As outlined in line 200, a lower $\rho$ (chosen as 0.8 in our experiments) enriches the shared structure's density while sparing the individual parts. Our preliminary investigations, spanning $\rho$ values from 0.1 to 0.9, indicated a stable predictive accuracy across the spectrum. We plan to enrich the appendix with these findings to demonstrate $\rho$'s effect on model performance comprehensively.

$\lambda$   This hyperparameter dictates the overall sparsity of both the shared and individual graph structures, with a higher value fostering sparser outcomes. Inspired by some previous researches[1, 2], we adopt $\lambda = \omega \sqrt{\frac{\log(Kp)}{n_{\text{tot}}}}$, where $\omega$ varies across $0.05 \times i | i = 1, 2, \ldots, 30$. The Bayesian Information Criterion (BIC) assists in pinpointing the optimal $\lambda$ for our model and is uniformly applied across baseline comparisons.

$\eta$   Regarding $\eta$, functioning as the soft-thresholding parameter, $\eta$'s selection leans on an empirical estimation of the graph sparsity by the user. It aims to adjust the number of nonzero entries in the model's sparsity pattern, correlating closely with those in the estimated precision matrix. A higher value of $\eta$ corresponds to a more sparsely estimated graph structure.

## B.7 Tables

Table 7 & 8 & 9 & 10 present the detaild of Fig. 3. They correspond to Subfigure (a)(b)(c)(d) correspondingly, showing time cost (seconds) comparison between FasMe and other baselines on simulated datasets, with dimension $p$ changing from 500 to 2500.

Table 7: The time comparisons of FasMe and the baselines on the simulated datasets varying $p$ and sample size $n = \mathbf{p}/\mathbf{20}$.

| Dimension($p$) | 500 | 1000 | 1500 | 2000 | 2500 |
|---|---|---|---|---|---|
| QUIC | 0.33 | 3.03 | 11.38 | 64.06 | 84.50 |
| Neighborhood Selection | 3.00 | 8.72 | 18.92 | 43.00 | 76.46 |
| Meta-IE | 4.33 | 27.27 | 89.23 | 96.85 | 112.63 |
| FasMe | **0.002** | **0.06** | **0.79** | **0.99** | **1.62** |

Table 8: The time comparisons of FasMe and the baselines on the simulated datasets varying $p$ and sample size $n = \mathbf{p}/\mathbf{10}$.

| Dimension($p$) | 500 | 1000 | 1500 | 2000 | 2500 |
|---|---|---|---|---|---|
| QUIC | 0.31 | 3.68 | 9.74 | 22.93 | 36.77 |
| Neighborhood Selection | 2.93 | 10.83 | 58.03 | 87.19 | 170.83 |
| Meta-IE | 2.62 | 16.66 | 39.76 | 87.14 | 96.11 |
| FasMe | **0.002** | **0.06** | **0.85** | **0.965** | **1.404** |

Table 9: The time comparisons of FasMe and the baselines on the simulated datasets varying $p$ and sample size $n = \mathbf{p}/\mathbf{5}$.

| Dimension($p$) | 500 | 1000 | 1500 | 2000 | 2500 |
|---|---|---|---|---|---|
| QUIC | 0.30 | 3.93 | 11.40 | 26.17 | 46.51 |
| Neighborhood Selection | 6.17 | 35.70 | 96.74 | 182.92 | 828.97 |
| Meta-IE | 1.65 | 9.90 | 10.29 | 14.08 | 14.10 |
| FasMe | **0.002** | **0.06** | **0.96** | **0.99** | **1.54** |

Table 10: The time comparisons of FasMe and the baselines on the simulated datasets varying $p$ and sample size $n = \mathbf{p}$.

| Dimension($p$) | 500 | 1000 | 1500 | 2000 | 2500 |
|---|---|---|---|---|---|
| QUIC | 0.30 | 3.56 | 8.38 | 22.37 | 39.54 |
| Neighborhood Selection | 13.56 | 84.21 | 216.44 | 402.84 | N/A |
| Meta-IE | 0.64 | 6.90 | 7.29 | 9.01 | 9.10 |
| FasMe | **0.001** | **0.05** | **0.68** | **0.97** | **1.34** |

# C  The full Algorithm of FasMe

---

**Algorithm 1** FasMe

---

1: **Input:** Auxiliary datasets $\{\mathbf{X}^{(k)}\}$, New task $\mathbf{X}_{\text{new}}$, hyper-parameter $\lambda$, $\rho$, $\gamma$, $\eta$, $c$, $\alpha$, the maximum iteration $T$, two thresholds $\epsilon_1$ and $\epsilon_2$, and a linear programming solver $\text{LP}(\cdot)$ that solves (18).

2: **for** $k = 1$ to $K$ **do**

3:      Calculate the covariance matrices $\Sigma^{(k)} = \frac{1}{n_k} \sum_{i=1}^{n_k} (\mathbf{X} - \bar{\mathbf{X}})(\mathbf{X} - \bar{\mathbf{X}})^\top$

4:      Initialize $\theta = \theta_r^{(k)} = \mathbf{0}_{p \times p}$

5:      Initialize $\mathbf{B}^{(k)} = \left[\mathbf{0}, \ldots, \Sigma^{(k)}, \ldots, \mathbf{0}, \frac{1}{\epsilon K}\Sigma^{(k)}\right]$

6: **end for**

7: **for** $i = 1$ to $p$ **do**

8:      $\mathbf{b} = \mathbf{e}_i, \boldsymbol{\beta} = \text{LP}(\mathbf{B}^{(k)}, \mathbf{b}), k = 1, \ldots, K$

9:      **for** $k = 1$ to $K$ **do**

10:         $\theta_{r,i}^{(k)} = \boldsymbol{\beta}_{(k-1)p+1:kp}$

11:      **end for**

12:      $\theta_i = \boldsymbol{\beta}_{(Kp+1):(K+1)p}$

13: **end for**

14: Calculate the sparsity pattern $\mathcal{G}$ by (8) and its chordal embedding $\tilde{\mathcal{G}}$ by Cholesky factorization

15: Initialize $y_1 = 0$

16: **for** $t = 1$ to $T$ **do**

17:      $\Delta y = -\nabla^2 l(y_t)^{-1} \nabla l(y_t)$

18:      $\alpha = 1$

19:      **if** $l(y + \alpha \Delta y) > l(y) + \gamma \alpha \Delta y^\top \nabla l(y)$ **then**

20:         $\alpha := \alpha \cdot c$

21:      **end if**

22:      Calculate the Newton decrement $\delta_k = |\Delta y_t^\top \nabla l(y_t)|$

23:      **if** $\delta_t < \epsilon_1$ or $|\Delta y_t| < \epsilon_2$ **then**

24:         Break

25:      **else**

26:         $y_{t+1} = y_t + \alpha \Delta y_t$

27:      **end if**

28: **end for**

29: **Output:** Precision matrix $\Omega_{\text{new}}$.

---

Similar to CLIME, (5) can also be solved column by column:

$$\left(\hat{\beta}, \{\hat{\beta}_r^{(k)}\}\right) = \underset{\beta, \{\beta_r^{(k)}\}}{\arg\min} \quad K\|\beta\|_1 + \rho \sum_{k=1}^{K} \|\beta_r^{(k)}\|_1,$$

$$\text{s.t.} \quad \|\beta\Sigma^{(k)} - (\mathbf{e}_j - \beta_r^{(k)}\Sigma^{(k)})\|_\infty \leq \lambda, \tag{16}$$

where $k = 1, 2, \ldots, K$. Here $\beta$ represents the corresponding column vector in the meta-knowledge $\theta$ and $\beta_r^{(k)}$ represents the $k$-th column vector in the rest $\theta_r^{(k)}$.

To solve (16), we can rewrite it as the following

$$\hat{\boldsymbol{\beta}} = \underset{\boldsymbol{\beta}}{\arg\min} \quad \|\boldsymbol{\beta}\|_1,$$

$$\text{s.t.} \quad \|\mathbf{B}^{(k)}\boldsymbol{\beta} - \mathbf{b}\|_\infty \leq \lambda, k = 1, \ldots, K. \tag{17}$$

Here, $\mathbf{B}^{(k)} = \left[\mathbf{0}, \ldots, \Sigma^{(k)}, \ldots, \mathbf{0}, \frac{1}{\epsilon K}\Sigma^{(k)}\right], \boldsymbol{\beta} = \left[(\beta^{(1)})^\top, \ldots, (\beta^{(k)})^\top, \ldots, (\beta^{(K)})^\top, \epsilon K(\beta^W)^\top\right]^\top$ and $\mathbf{b} = \mathbf{e}_j$.

Through relaxation, we can convert (17) to the following linear programming formulation:

$$\hat{\mathbf{a}}_j = \underset{\mathbf{a}_j}{\arg\min} \sum_{j=1}^{(K+1)p} \mathbf{a}_j,$$

$$\text{s.t.} \quad -\theta_j \leq \mathbf{a}_j, \ j = 1, \ldots, (K+1)p,$$
$$\theta_j \leq \mathbf{a}_j, \ j = 1, \ldots, (K+1)p, \tag{18}$$
$$-(\mathbf{B}_i^{(k)})^\top \theta + \mathbf{b}_i \leq c, i = 1, \ldots, p; k = 1, \ldots, K,$$
$$(\mathbf{B}_i^{(k)})^\top \theta - \mathbf{b}_i \leq c, i = 1, \ldots, p; k = 1, \ldots, K.$$

Here $\mathbf{a}_j$ are the slack variables. $\mathbf{B}_i^{(k)}$ represents the $i$-th row of $\mathbf{B}^{(k)}$ and $\mathbf{b}_i$ is the $i$-th entry of $\mathbf{b}$. According to [25], we can apply the same symmetric operators on $\Omega^{(k)} = \theta + \theta_r^{(k)}$ obtained from Algorithm 1.

Then we combined the support set of the thresholded covariance and meta-knowledge to obtain the sparsity pattern:

$$\mathcal{G} = \text{supp}(\theta) \cup \text{supp}\left(S_\eta(\Sigma_{\text{new}})\right). \tag{19}$$

To solve the optimization problem (9), we first need to obtain the chordal embedding $\tilde{\mathcal{G}}$ by Cholesky factorization. We compute the unique lower triangular Cholesky factor $L$ satisfying $\Pi_{\mathcal{G}}(\Sigma_{\text{new}}) = LL^\top$. After ignoring perfect numerical cancellation, we have the adjacency matrix of the chordal embedding $L + L^\top$ using symbolic Cholesky algorithm.

We define the cone of sparse positive semidefinite matrices $\mathcal{P}$, and the cone of sparse matrices with positive semidefinite completions $\mathcal{P}_*$ as the following:

$$\mathcal{P} = \mathbb{S}_+^p \cap \mathbb{S}_{\tilde{\mathcal{G}}}^p, \quad \mathcal{P}_* = \{\langle S, \Omega \rangle \geq 0 : S \in \mathbb{S}_{\tilde{\mathcal{G}}}^p\}.$$

The primal and dual problem of Eq. (9) can be written as:

$$\underset{\Omega \in \mathcal{P}}{\arg\min} \langle \Pi_{\mathcal{G}}(\Sigma), \Omega \rangle + g(\Omega) \quad \text{s.t. } P^\top(\Omega) = 0, \tag{20}$$

$$\underset{S \in \mathcal{P}, y \in \mathbb{R}^m}{\arg\max} -g_*(S) \quad \text{s.t. } S = \hat{\Sigma} - P(y), \tag{21}$$

where $P(\cdot) : \mathbb{R}^m \to \mathbb{S}_{\tilde{\mathcal{G}} \backslash \mathcal{G}}^p$ represents a linear map that converts a list of $m$ variables into the corresponding matrix in $\tilde{\mathcal{G}} \backslash \mathcal{G}$. $g$ and $g_*$ are the "log-det" barrier functions on $\mathcal{P}$ and $\mathcal{P}_*$:

$$g(\Omega) = -\log \det \Omega, \quad g_*(S) = -\min_{\Omega \in \mathcal{P}} \langle S, \Omega \rangle - \log \det \Omega.$$

Under the sparsity and chordal assumption, the gradient evaluations and Hessian matrix-vector products can be efficiently evaluated in $O(p)$ time and $O(p)$ memory, using the numerical recipes described in [52].

To solve the dual problem (21), we rewrite it into the following unconstrained optimization problem:

$$\hat{y} \equiv \underset{y \in \mathbb{R}^m}{\arg\min} \, l(y) \equiv g_*(\Pi_{\mathcal{G}}(\Sigma_{\text{new}}) - P(y)). \tag{22}$$

After obtaining the solution $\hat{y}$, we can recover the optimal estimator for the primal problem through $\hat{\Omega} = -\nabla g_*(\Pi_{\mathcal{G}}(\Sigma_{\text{new}}) - P(y))$. The dual problem is easier to solve than the primal because the initial point $y = 0$ tends to lie close to the solution $\hat{y}$. Starting from this point, we can use Newton's method to converge rapidly.

We then solve the Newton direction $\Delta y$ via the $m \times m$ system of equations

$$\nabla^2 l(y) \Delta y = -\nabla l(y). \tag{23}$$

Starting from the origin $y_1 = 0$, the method converges to an $\epsilon$-accurate search direction $y_1$ satisfying

$$(y_1 - \Delta y)^\top \nabla^2 l(y)(y_1 - \Delta y) \leq \epsilon |\Delta y^\top \nabla l(y)| \tag{24}$$

in at most $\sqrt{\|\nabla^2 l(y)\| \|\nabla^2 l(y)^{-1}\|} \log(\frac{2}{\epsilon})$ conjugate gradient iterations.

**Computational Complexity** In this section, we mainly analyze the time and memory complexity of our estimator in the meta-testing step. The time and memory cost of our estimator can be divided into two parts. The first part is to threshold the covariance matrix. This step is quadratic $O(p^2)$ time and memory but embarrassingly parallelizable. The second part is to solve (7) using Newton Conjugate Gradient methods. If the prior sparsity pattern $\mathcal{G}$ is sparse and chordal, then the second part can be performed using our algorithm in linear $O(p)$ time and memory. This significantly outperforms QUIC [10], Neighborhood Selection method [34] and Bigquic used in [11].

## D   Limitations

As discussed earlier, the biological and genetic domains have adopted and validated the common structure assumption based on some biological evidence. For other domains, if the edges are sparse and the data are high-dimensional, it may be difficult to recover the common structure from the data without some prior evidence. In the worst case, the meta-teacher can learn the knowledge that there is no common edge among the tasks. The meta-student has to learn from scratch. The optimization problem in the meta-testing process can be seen as a graphical lasso problem (introduced in Section 2.2). The neighborhood selection baseline is one type of algorithm that solves the graphical lasso. Consequently, our proposed method remains at least as good as the baseline, if not stronger, in handling such challenges.

## E   Explanations about the Choices of the Baselines

We chose QUIC and Neighborhood Selection since they continue to be widely utilized in current research and serve as critical benchmarks for new methodologies within the realm of state-of-the-art works.

We take QUIC as an example. [53] utilizes the QUIC method to detect structural changes in high-dimensional Gaussian graphical models. QUIC's ability to perform fast and accurate estimation underpins the methodology for identifying change-points in the graphical model structure over time. Similarly, studies like those in recent works like [54, 55] also employ QUIC for various analytical tasks downstream.

Additionally, while both [56] and [57] utilize QUIC as a baseline for comparison, [56] relies on a structural assumption of Kronecker-sum-structured inverse covariance, and [57] operates under a general missing dependency assumption. These specific premises limit their applicability, rendering them less generalizable to our scenario.

Parallel to QUIC's applicability, Neighborhood Selection is similarly employed in contemporary studies, demonstrating its ongoing relevance and utility [58, 59, 60, 61].

As mentioned above, we observed two main categories. The first category of recent works did not propose improvements to QUIC; instead, they applied it to more specific tasks. The second category, while outperforming QUIC as a baseline and demonstrating improvements, often relied on stricter structural assumptions not compatible with our problem context. Consequently, both categories were not suitable as baselines for our study.

We specifically chose Meta-IE method due to its closeness to our work. Meta-IE represents the latest advancement in meta-learning applied to sparse precision matrix estimation, making it an ideal candidate for direct comparison.

To this end, we have decided to incorporate an evaluation against gRankLasso [35], a recent development in graph learning, into our revised manuscript. With a completely tuning-free technique, gRankLasso shows better accuracy performance than CLIME, GLasso, and TIGER. However, its extensive time consumption for hyperparameter simulations and rank calculations poses a drawback, particularly underperforming with small datasets.

## F   Proof of Theorems

**Definition 2.** *Let* $\mathbf{X}_1^{(k)}, \ldots, \mathbf{X}_{n_k}^{(k)} \in \mathbb{R}^p$ *be i.i.d. random vectors for* $1 \leq k \leq K$. *Let* $\mathbf{X}_{ij}$ *be the* $j$-*th entry of* $\mathbf{X}_i^{(k)}$ *for* $1 \leq j \leq p$. *We use* $(\Sigma^{(k)})^*, (\Omega^{(k)})^*, \theta^*$ *to represent the real covariance*

matrix, precision matrix, and meta-knowledge, respectively. $\{\mathbf{X}^{(k)}\}$ follows a family of random $p$-dimensional multivariate sub-Gaussian distributions with parameter $\sigma$ if

(1) $\left\{\mathbf{X}_i^{(k)}\right\}_{1 \le i \le n_k, 1 \le k \le K}$ are conditionally independent given $\{(\Omega^{(k)})^*\}_{k=1}^K$;

(2) $\dfrac{\mathbf{X}_{ij}^{(k)}}{\sqrt{\sigma_{jj}^{(k)}}}$ conditioned on $(\Omega^{(k)})^*$ is sub-Gaussian with parameter $\sigma$ for $1 \le i \le n_k, 1 \le j \le p, 1 \le k \le K$;

(3) $\mathbb{E}[\mathbf{X}_i^{(k)}|(\Sigma^{(k)})^*] = 0, \mathrm{Cov}\left(\mathbf{X}_i^{(k)}|(\Sigma^{(k)})^*\right) = \left(\Sigma^{(k)}\right)^* = \left((\Omega^{(k)})^*\right)^{-1}$ for $1 \le i \le n_k, 1 \le k \le K$.

Then we remove the assumption on the support set of $\theta_r^{(k)}$, thus relaxing the data assumption of [11]:

(4) $(\Omega^k)^* = \theta^* + \theta_r^{(k)}$ with $\theta, \theta_r^{(k)} \in \mathbb{R}^{p \times p}$. The meta-knowledge $\theta$ is deterministic, and $\theta_r^{(k)}, 1 \le k \le K$, are i.i.d. random matrices.

**Lemma 1.** *Given a simple optimization problem, $\hat{x}$ and $\hat{y}$ are the optimal solution.*

$$\hat{x} + \hat{y} = \arg\min_{x,y} |x| + \rho|y|$$
$$\text{s.t. } x + y = C \tag{25}$$

*where $\rho > 0$ and $C$ is a constant. Then we have $\hat{x} \cdot \hat{y} \ge 0$.*

*Proof.* If $\hat{x} \cdot \hat{y} < 0$, let $x' = \hat{x} + \hat{y}$ and $y' = 0$. Without loss of generality, we assume that $|\hat{x}| \ge |\hat{y}|$. It follows that

$$|x'| + \rho|y'| = |\hat{x} + \hat{y}| < |\hat{x}| < |\hat{x}| + \rho|\hat{y}|, \tag{26}$$

which means that the pair $x', y'$ is a better solution. This contradicts the earlier stated premise that $\hat{x}, \hat{y}$ are the optimal solution. The proof is completed by contradiction.

**Lemma 2.** *If each $\theta_r^{(k)} + \theta$ satisfies (10), then $\Omega_{\mathrm{tot}}$ also satisfies (10). Thus $\hat{\Omega}_{\mathrm{tot}}$ satisfies the condition $\hat{\Omega}_{\mathrm{tot}} \succ 0$, with a high probability.*

**Corollary 1.** *We assume that $\hat{\theta}$ is the optimal solution of Eq. (5) and $\hat{\theta}_r^{(k)} = \hat{\Omega}^{(k)} - \hat{\theta}$. Based on Lemma 1, it follows that $\hat{\theta}_{r,ij}^{(k)} \cdot \hat{\theta}_{ij} \ge 0$. Then it is simple to see that*

$$\|\hat{\Theta}_r\|_\infty \le \|\hat{\Theta}_r\|_\infty + \|\hat{\Theta}\|_\infty = \|\hat{\Theta}_r + \hat{\Theta}\|_\infty = \|\hat{\Omega}_{\mathrm{tot}}\|_\infty. \tag{27}$$

**Theorem 1.** *We use $\hat{\Theta}, \Theta^*$ to represent the estimated precision matrix and real precision matrix, respectively. Suppose that $\Omega_{\mathrm{tot}}^* \in \mathcal{U}$. If $\lambda \ge \nu(\max_{ij} |\hat{\sigma}_{ij} - \sigma_{ij}^*|)$, we have that*

$$\|\hat{\Theta} - \Theta^*\|_\infty \le 8\nu\lambda + 2\phi. \tag{28}$$

*where $\nu = \|\Omega_{\mathrm{tot}}^*\|_1$ and $\phi = \|\Omega_{\mathrm{tot}}^*\|_\infty$.*

*Proof.* According to the condition in Theorem 1,

$$\|\hat{\Sigma}_{\mathrm{tot}} - \Sigma_{\mathrm{tot}}^*\|_\infty \le \frac{\lambda}{\|\Omega_{\mathrm{tot}}^*\|_1}. \tag{29}$$

Then we can show that

$$\|I - \hat{\Sigma}_{\mathrm{tot}} \Omega_{\mathrm{tot}}^*\|_\infty = \|(\Sigma_{\mathrm{tot}}^* - \hat{\Sigma}_{\mathrm{tot}})\Omega_{\mathrm{tot}}^*\|_\infty \le \|\Omega_{\mathrm{tot}}^*\|_1 \|\Sigma_{\mathrm{tot}}^* - \hat{\Sigma}_{\mathrm{tot}}\|_\infty \le \lambda, \tag{30}$$

where $|\mathbf{A}\mathbf{B}|_\infty \le |\mathbf{A}|_\infty |\mathbf{B}|_1$ for any matrices $\mathbf{A}, \mathbf{B}$ of appropriate sizes.

Then we can rewrite (5) as:

$$\left(\hat{\Theta}_r, \hat{\Omega}_{\mathrm{tot}}\right) = \arg\min_{\{\theta_r^{(k)}\}, \Omega_{\mathrm{tot}}} \|\Omega_{\mathrm{tot}}\|_1 + (1-\rho)\|\Theta_r\|_1$$
$$\text{s.t. } \|\Sigma_{\mathrm{tot}}\Omega_{\mathrm{tot}} - I\|_\infty \le \lambda. \tag{31}$$

This is because $\|\Omega_{\mathrm{tot}}\|_1 + (1-\rho)\sum_{k=1}^K \|\theta_r^{(k)}\|_1 = \sum_{k=1}^K \|\theta + \theta_r^{(k)}\|_1 + (1-\rho)\sum_{k=1}^K \|\theta_r^{(k)}\|_1$. Notice that the constraint is not related to $\Theta_r$. Therefore $\hat{\Omega}_{\mathrm{tot}}$ is also the solution of the following optimization problem:

$$\hat{\Omega}_{\mathrm{tot}} = \arg\min_{\Omega_{\mathrm{tot}}} \|\Omega_{\mathrm{tot}}\|_1 \quad \text{s.t.} \|\Sigma_{\mathrm{tot}}\Omega_{\mathrm{tot}} - I\|_\infty \le \lambda. \tag{32}$$

Since $\Omega^*_{\text{tot}}$ and $\hat{\Omega}_{\text{tot}}$ satisfy (32), we have

$$\|\hat{\Omega}_{\text{tot}}\|_1 \leq \|\Omega^*_{\text{tot}}\|_1. \tag{33}$$

It suggests that

$$\|\hat{\Sigma}_{\text{tot}}(\hat{\Omega}_{\text{tot}} - \Omega^*_{\text{tot}})\|_\infty \leq \|\hat{\Sigma}_{\text{tot}}\hat{\Omega}_{\text{tot}} - I\|_\infty + \|I - \hat{\Sigma}_{\text{tot}}\Omega^*_{\text{tot}}\|_\infty \leq 2\lambda. \tag{34}$$

Consequently, it is straightforward to show that

$$\begin{aligned}
\|\Sigma^*_{\text{tot}}(\hat{\Omega}_{\text{tot}} - \Omega^*_{\text{tot}})\|_\infty \leq & \|\hat{\Sigma}_{\text{tot}}(\hat{\Omega}_{\text{tot}} - \Omega^*_{\text{tot}})\|_1 + \|(\hat{\Sigma}_{\text{tot}} - \Sigma^*_{\text{tot}})(\hat{\Omega}_{\text{tot}} - \Omega^*_{\text{tot}})\|_\infty \\
\leq & 2\lambda + \|\hat{\Omega}_{\text{tot}} - \Omega^*_{\text{tot}}\|_1 \|\hat{\Sigma}_{\text{tot}} - \Sigma^*_{\text{tot}}\|_\infty \leq 4\lambda.
\end{aligned} \tag{35}$$

Then, it follows that

$$\|\hat{\Omega}_{\text{tot}} - \Omega^*_{\text{tot}}\|_\infty \leq \|\Omega^*_{\text{tot}}\|_1 \|\Sigma^*_{\text{tot}}(\hat{\Omega}_{\text{tot}} - \Omega^*_{\text{tot}})\|_\infty \leq 4\|\Omega^*_{\text{tot}}\|_1 \lambda = 4\nu\lambda, \tag{36}$$

where $\|\Omega^*_{\text{tot}}\|_1 = \nu$. Based on the triangle inequality for norm, we have that

$$\begin{aligned}
\|\hat{\Theta} - \Theta^*\|_\infty &= \|\hat{\Omega}_{\text{tot}} - \hat{\Theta}_r - (\Omega^*_{\text{tot}} - \Theta^*_r)\|_\infty \\
&\leq \|\hat{\Omega}_{\text{tot}} - \Omega^*_{\text{tot}}\|_\infty + \|\hat{\Theta}_r - \Theta^*_r\|_\infty \\
&\leq \|\hat{\Omega}_{\text{tot}} - \Omega^*_{\text{tot}}\|_\infty + \|\hat{\Theta}_r\|_\infty + \|\Theta^*_r\|_\infty.
\end{aligned} \tag{37}$$

In terms of $\Theta^*, \Theta^*_r$, we assume $W^*_{\text{tot},ij} \cdot \Omega^*_{I,ij} = 0$ and $W^*_{\text{tot},ij} + \Omega^*_{I,ij} = 0$ *iff* $W^*_{\text{tot},ij} = \Omega^*_{I,ij} = 0$. Thus we can derive that $\|\Theta^*\|_\infty \leq \|\Omega^*_{\text{tot}}\|_\infty$. We are now turning to the proof of Eq. (37). Combining Corollary 1 and Theorem 1, we can show that

$$\begin{aligned}
\|\hat{\Theta} - \Theta^*\|_\infty &\leq \|\hat{\Omega}_{\text{tot}} - \Omega^*_{\text{tot}}\|_\infty + \|\hat{\Omega}_{\text{tot}}\|_\infty + \|\Omega^*_{\text{tot}}\|_\infty \\
&\leq \|\hat{\Omega}_{\text{tot}} - \Omega^*_{\text{tot}}\|_\infty + \|\hat{\Omega}_{\text{tot}} - \Omega^*_{\text{tot}}\|_\infty + 2\|\Omega^*_{\text{tot}}\|_\infty \\
&\leq 2\|\hat{\Omega}_{\text{tot}} - \Omega^*_{\text{tot}}\|_\infty + 2\|\Omega^*_{\text{tot}}\|_\infty \\
&\leq 8\|\Omega^*_{\text{tot}}\|_1 \lambda + 2\|\Omega^*_{\text{tot}}\|_\infty \\
&= 8\nu\lambda + 2\phi,
\end{aligned} \tag{38}$$

where $\|\Omega^*_{\text{tot}}\|_\infty = \phi$. The proof is completed.

**Theorem 2.** *Suppose that $\Omega^*_{\text{tot}} \in \mathcal{U}$ and $N = \sum_{k=1}^K n_k$. When the variables are sub-Gaussian, the tail condition holds. Let $\lambda = C_0 \nu \sqrt{\frac{\log(Kp)}{N}}$, where the constant $C_0 = 2\gamma^{-2}(2 + \tau_0 + \gamma^{-1}e^2C^2)^2$ and constant $\tau_0 > 0$. If $\omega^{\text{tot}}_{\min} > 2\tau_n$, then we have*

1. *$\|\hat{\theta} - \theta^*\|_\infty \leq 8C_0\nu^2 \sqrt{\frac{\log(Kp)}{N}} + 2\phi$;*

2. *$\text{supp}(\tilde{\theta}) = \text{supp}(\theta^*)$;*

*with a probability greater than $1 - 4p^{-\tau_0}$. Here $\tilde{\theta}$ is a threshold estimator with $\tilde{\theta}_{ij} = \hat{\theta}_{ij} I(|\hat{\theta}_{ij}| \geq \tau_n)$, where $\tau_n \geq 4\nu\lambda$ is a tuning parameter.*

*Proof.* To obtain the first conclusion of Theorem 2, we need to prove

$$\max_{ij} |\hat{\sigma}_{ij} - \sigma^*_{ij}| \leq C_0 \sqrt{\frac{\log Kp}{N}} \tag{39}$$

with a probability greater than $1 - 4p^{-\tau_0}$ under the exponential tail condition. Without loss of generality, we assume that $\mathbb{E}(\mathbf{X}^{(k)}) = 0$. Let $\Sigma'_{\text{tot}} := N^{-1}\sum_{k=1}^N \mathbf{X}_{\text{tot},k}\mathbf{X}^\top_{\text{tot},k}$ and $\mathbf{Y}_{kij} = \mathbf{X}_{\text{tot},ki}\mathbf{X}_{\text{tot},kj} - \mathbb{E}(\mathbf{X}_{\text{tot},ki}\mathbf{X}_{\text{tot},kj})$. We then have $\hat{\Sigma}_{\text{tot}} = \Sigma'_{\text{tot}} - \bar{\mathbf{X}}_{\text{tot}}\bar{\mathbf{X}}^\top_{\text{tot}}$. Let $t = \gamma\sqrt{\log Kp/N}$. Using the inequality $|e^s - 1 - s| < s^2 e^{\max(s,0)}$ for any $s \in \mathbb{R}$ and let $C_1 = 2 + \tau_0 + \gamma^{-1}C^2$. Then by some basic calculations, we can get

$$\begin{aligned}
\mathbb{P}\left(\sum_{k=1}^N \mathbf{Y}_{kij} \geq \gamma^{-1}C_1\sqrt{N\log Kp}\right) &\leq \exp(-C_1 \log Kp)(\mathbb{E}[\exp(t\mathbf{Y}_{kij})])^N \\
&\leq \exp(-C_1 \log Kp + Nt^2\mathbb{E}Y^2_{kij}\exp(t|\mathbf{Y}_{kij}|)) \\
&\leq \exp(-C_1 \log Kp + \gamma^{-1}C^2 \log Kd) \\
&\leq \exp(-(\tau_0 + 2)\log Kp).
\end{aligned} \tag{40}$$

Then we obtain

$$\mathbb{P}\left(\|\Sigma'_{\text{tot}} - \Sigma^*_{\text{tot}}\|_\infty \geq \gamma^{-1} C_1 \sqrt{\frac{\log Kp}{N}}\right) \leq 2p^{-\tau_0}. \tag{41}$$

Using the inequality $e^s \leq e^{s^2+1}$ for $s > 0$, we have

$$\mathbb{E}e^{t|X_j|} \leq eC, \ \forall t \leq \sqrt{\gamma}. \tag{42}$$

Let

$$C_2 = 2 + \tau_0 + \gamma^{-1} e^2 C^2 \tag{43}$$

and $a_n = C_2^2 (\log Kp/N)^{\frac{1}{2}}$. As before, we can have that

$$\mathbb{P}\left(\|\bar{\mathbf{X}}_{\text{tot}}\bar{\mathbf{X}}_{\text{tot}}^\top\|_\infty \geq \gamma^{-2} a_n \sqrt{\log Kp/N}\right) \leq p \max_i \mathbb{P}\left(\sum_{k=1}^N \mathbf{X}_{ki} \geq \gamma^{-1} C_2 \sqrt{N \log Kp}\right)$$
$$+ p \max_i \mathbb{P}\left(-\sum_{k=1}^N \mathbf{X}_{ki} \geq \gamma^{-1} C_2 \sqrt{N \log Kp}\right)$$
$$\leq 2p^{-\tau_0 - 1} \tag{44}$$

By (41), (44), and the inequality $C > \gamma^{-1} C_1 + \gamma^{-2} a_n$, we see that (39) holds. The proof of the first inequality is completed.

To prove the sign-consistency of $\hat{\Theta}$, we define a threshold estimator $\tilde{\Omega}_{\text{tot}}$ with $\tilde{\omega}_{ij} = \hat{\omega}_{ij} I\{|\hat{\omega}_{ij}| \geq \tau_n\}$ where $\tau_n \geq 4\nu\lambda$ is a tuning parameter. We define $\omega_{\min}^{\text{tot}} = \min_{(i,j)\in\text{supp}(\Omega^*_{\text{tot}})} |\Omega^*_{\text{tot,ij}}|$ and a threshold estimator $\tilde{\Theta}$ with $\tilde{\Theta}_{ij} = \hat{\Theta}_{ij} I\{|\hat{\Theta}_{ij}| \geq \tau_n\}$, where $\tau \geq 4\nu\lambda$. It is known that the resulting elements in $\hat{\Omega}_{\text{tot}}$ will exceed the threshold level if the corresponding element in $\Omega^*_{\text{tot}}$ is large in magnitude. In contrast, the entries of $\hat{\Omega}_{\text{tot}}$ outside the support of $\Omega^*_{\text{tot}}$ will remain below the threshold level with high probability. As a result, if $\omega_{\min}^{\text{tot}} > 2\tau_n$, we have $\text{supp}(\tilde{\Omega}_{\text{tot}}) = \text{supp}(\Omega^*_{\text{tot}})$. Since the nonzero entries in $\theta$ correspond to the intersection of the nonzero entries in $\{\Omega^{(k)}\}$, it is straightforward to get that $\text{supp}(\tilde{\Theta}) = \text{supp}(\Theta^*)$ if $\omega_{\min}^{\text{tot}} > 2\tau_n$.

**Lemma 3.** *Consider a zero-mean random vector $(\mathbf{X}_1, \ldots, \mathbf{X}_p)$ with covariance $\Sigma^*$ such that each $\frac{\mathbf{X}_i}{\sqrt{\Sigma^*_{ii}}}$ is sub-Gaussian with parameter $\sigma$. We define $W = \hat{\Sigma}_{\text{new}} - \Sigma^*_{\text{new}}$. Given $n$ i.i.d. samples, the associated sample covariance $\hat{\Sigma}$ satisfies the tail bound if $\|\Sigma^*_{\text{new}}\|_\infty \leq \varphi$:*

$$\mathbb{P}[\|W\|_\infty > \delta] \leq 4|\mathcal{G}| \exp\left\{-\frac{n\delta^2}{128(1+4\sigma^2)^2\varphi^2}\right\} \tag{45}$$

*for all $\delta \in (0, 8\varphi(1 + 4\sigma^2))$.*

**Theorem 3.** *Suppose we have recovered the true sparsity pattern $\mathcal{G}$ of a family of $d$-dimensional random multivariate sub-Gaussian distribution. Then for a new task of multivariate sub-Gaussian distribution with the precision matrix $\Omega^*_{\text{new}}$ such that $\text{supp}(\theta) \subseteq \text{supp}(\Omega^*_{\text{new}})$ and satisfying irrepresentable condition, consider the estimator $\hat{\Omega}_{\text{new}}$ with $\lambda = C_3\nu\sqrt{\frac{\log|\mathcal{G}|}{n}}$. Then we have*

$$\|\hat{\Omega}_{\text{new}} - \Omega^*_{\text{new}}\|_\infty \leq 2C_3\nu^2\kappa_{\Gamma^*}\sqrt{\frac{\log|\mathcal{G}|}{n}}, \tag{46}$$

*with a probability $1 - p^{-\tau_1}$, where $\tau_1 > 0$.*

The proof is similar to Theorem 2 and will be omitted.

**Definition 3.** *[32] Given a matrix $M \in \mathbb{S}^p$, define $\mathcal{G}_M = \{(i, j) : M_{ij} \neq 0$ as its sparsity pattern. Then $M$ is called **inverse-consistent** if there exists a matrix $N \in \mathbb{S}^p$ such that*

$$M + N \succeq 0$$
$$N = 0 \quad \forall(i, j) \in \mathcal{G}_M \tag{47}$$
$$(M + N)^{-1} \in \mathbb{S}^p_{\mathcal{G}_M}$$

The matrix $N$ is called an inverse-consistent complement of $M$ and is denoted by $M^{(c)}$. Furthermore, $M$ is called **sign-consistent** if for every $(i,j) \in \mathcal{G}_M$, the $(i,j)$-th elements of $M$ and $(M + M^{(c)})^{-1}$ have opposite signs.

Then we define the $\beta(\mathcal{G}, \alpha)$ function defined with respect to the sparsity pattern $\mathcal{G}$ and scalar $\alpha > 0$

$$
\begin{aligned}
\beta(\mathcal{G}, \alpha) = \max_{M \succ 0} \quad & \|M^{(c)}\|_{\max} \\
\text{s.t.} \quad & M \in \mathbb{S}_{\mathcal{G}}^p \text{ and } \|M\|_{\max} \leq \alpha \\
& M_{i,i} = 1 \quad \forall i \in \{1, \ldots, p\} \\
& M \text{ is inverse-consistent.}
\end{aligned}
$$

**Lemma 4.** *[32] Any arbitrary matrix with positive-definite completion is inverse-consistent and has a unique inverse-consistent complement.*

**Lemma 5.** $\hat{\Omega}$ *is the optimal solution if and only if it satisfies the following conditions for every* $(i,j) \in \{1, \ldots, p\}^2$:

$$
\begin{aligned}
\bar{\Omega}_{ij}^{-1} &= \Sigma_{ij} \quad \text{if } i = j \\
\bar{\Omega}_{ij}^{-1} &= \Sigma_{ij} + \eta \times \text{sign}(\bar{\Omega}_{ij}) \quad \text{if } \bar{\Omega}_{ij} \neq 0 \\
\Sigma_{ij} - \eta &\leq \bar{\Omega}_{ij}^{-1} \leq \Sigma_{ij} + \eta.
\end{aligned}
\tag{48}
$$

*Proof.* The proof is straightforward and omitted for brevity.

Then, consider the following optimization problem:

$$
\min_{\Omega \in \mathbb{S}_+^d} - \log \det(\Omega) + \langle \tilde{\Sigma}, \Omega \rangle + \sum_{(i,j) \in \mathcal{G}} \tilde{\eta} |\Omega_{ij}| + 2 \max_k \Sigma_{kk} \sum_{(i,j) \in \mathcal{G}^c} |\Omega_{ij}|
\tag{49}
$$

where

$$
\tilde{\Sigma}_{ij} = \frac{\Sigma_{ij}}{\sqrt{\Sigma_{ii} \times \Sigma_{jj}}}, \quad \tilde{\eta} = \frac{\eta}{\sqrt{\Sigma_{ii} \times \Sigma_{jj}}}.
$$

We denote $\tilde{\Omega}$ as the optimal solution of (49) and define $D$ as a diagonal matrix with $D_{ii} = \Sigma_{ii}$ for every $i = 1, \ldots, p$.

**Lemma 6.** *We have* $\bar{\Omega} = D^{-\frac{1}{2}} \times \tilde{\Omega} \times D^{-\frac{1}{2}}$.

**Theorem 4.** $\mathcal{G}$ *coincides with the sparsity pattern of the optimal solution* $\Omega_{\text{new}}^*$ *if the normalized matrix* $\tilde{\Sigma}_{\text{new}} = D^{-1/2} \Pi_{\mathcal{G}}(\Sigma_{\text{new}}) D^{-1/2}$ *where* $D = \text{diag}(\Pi_{\mathcal{G}}(\Sigma_{\text{new}}))$ *satisfies the following conditions:*

1. $\tilde{\Sigma}_{\text{new}}$ *is positive definite,*

2. $\tilde{\Sigma}_{\text{new}}$ *is sign-consistent,*

3. *We have*

$$
\begin{aligned}
&\beta\left(\mathcal{G}, \|\tilde{\Sigma}_{\text{new}}\|_{\max}\right) \\
&\leq \min_{(k,l) \notin \mathcal{G}} \frac{\eta - |(\Pi_{\mathcal{G}}(\Sigma_{\text{new}}))_{kl}|}{\sqrt{(\Pi_{\mathcal{G}}(\Sigma_{\text{new}}))_{kk} \cdot (\Pi_{\mathcal{G}}(\Sigma_{\text{new}}))_{ll}}}.
\end{aligned}
\tag{50}
$$

*Proof.* We define an intermediate optimization problem as the following:

$$
\Omega^\dagger = \arg\min_{\Omega \in \mathbb{S}_+^p} f(\Omega) = -\log \det \Omega + \langle \Sigma, \Omega \rangle + \sum_{(i,j) \in \mathcal{G}} \lambda_{ij} |\Omega_{ij}| + 2 \max_k \Sigma_{kk} \sum_{(i,j) \in \mathcal{G}^c} |\Omega_{ij}|.
\tag{51}
$$

First, we show that $\Omega^\dagger = \bar{\Omega}$. Trivially, $\bar{\Omega}$ is a feasible solution for (51) and hence $f(\Omega^\dagger) \leq f(\bar{\Omega})$. Now, we prove that $\Omega^\dagger$ is a feasible solution. We show that $\Omega_{ij}^\dagger = 0$ for every $(i,j) \in \mathcal{G}^c$. By contradiction, suppose $\Omega_{ij}^\dagger \neq 0$ for some $(i,j) \in \mathcal{G}^c$. Note that, due to the positive definiteness of $(\Omega^\dagger)^{-1}$, we obtain

$$
(\Omega^\dagger)_{ii}^{-1} \times (\Omega^\dagger)_{jj}^{-1} - ((\Omega^\dagger)_{ij}^{-1})^2 > 0.
\tag{52}
$$

Then, based on Lemma 6, it follows that

$$(\Omega^\dagger)^{-1}_{ij} = \Sigma_{ij} + 2 \max_k \Sigma_{kk} \times \text{sign}(\Omega^\dagger_{ij}). \tag{53}$$

Considering the fact that $\Sigma \succ 0$, we have $|\Sigma_{ij}| \le \max_k \Sigma_{kk}$. It implies that $|(\Omega^\dagger)^{-1}_{ij}| \ge \max_k \Sigma_{kk}$. Furthermore, due to Lemma 5, we have $(\Omega^\dagger)^{-1}_{ii} = \Omega_{ii}$ and $(\Omega^\dagger)^{-1}_{jj} = \Omega_{jj}$. This leads to

$$\left(\Omega^\dagger\right)^{-1}_{ii} \times \left(\Omega^\dagger\right)^{-1}_{jj} - \left(\left(\Omega^\dagger\right)^{-1}_{ij}\right)^2 = \Sigma_{ii} \times \Sigma_{jj} - \left(\max_k \{\Sigma_{kk}\}\right)^2 \le 0, \tag{54}$$

which contradicts with (52). Therefore, $\Omega^\dagger$ is a feasible solution. This implies that $f(\Omega^\dagger) \ge f(\bar{\Omega})$ and hence, $f(\bar{\Omega}) = f(\Omega^\dagger)$. Due to the uniqueness of the solution of (51), we have $\bar{\Omega} = \Omega^\dagger$. Now, note that (51) can be reformulated as

$$\min_{\Omega \in \mathbb{S}^p_+} - \log \det(\Omega) + \langle \tilde{\Sigma}, D^{1/2}\Omega D^{1/2}\rangle + \sum_{(i,j)\in\mathcal{G}} \eta \, |\Omega_{ij}| + 2 \max_k \Sigma_{k,k} \sum_{(i,j)\in\mathcal{G}^c} |\Omega_{ij}|. \tag{55}$$

Upon defining

$$\tilde{\Omega} = D^{1/2}\Omega D^{1/2} \tag{56}$$

and following some algebra, one can verify that (51) is equivalent to

$$\min_{\tilde{\Omega} \in \mathbb{S}^d_+} - \log \det(\tilde{\Omega}) + \langle \tilde{\Sigma}, \tilde{\Omega}\rangle + \sum_{(i,j)\in\mathcal{G}} \tilde{\eta} \left|\tilde{\Omega}_{ij}\right| + 2 \max_k \left\{\tilde{\Sigma}_{kk}\right\} \sum_{(i,j)\in\mathcal{G}^{(c)}} \left|\tilde{\Omega}_{ij}\right| + \log \det(D). \tag{57}$$

Dropping the constant term in (57) gives rise to the (49). As a result, $\bar{\Omega} = D^{-\frac{1}{2}} \times \tilde{\Omega} \times D^{-\frac{1}{2}}$ holds. The proof is completed.

Next, we prove that the newton subproblem can be solved in $O(1)$ CG iteration. Let $\mathbb{S}^p$ be the set of $p \times p$ real symmetric matrices. Given a sparsity pattern $V$, we define $\mathbb{S}^p_V \subseteq \mathbb{S}$ as the set of $p \times p$ real symmetric matrices with this sparsity pattern. We consider the following minimization problem

$$\hat{y} \equiv \arg \min_{y \in \mathbb{R}^m} l(y) \equiv g_*(\Sigma - P(y)). \tag{58}$$

Here, the problem data $P : \mathbb{R}^m \to \mathbb{S}^p_F$ is an orthogonal basis for a sub-sparsity pattern $F \subset V$ that excludes the matrix $\Sigma$. In other words, the operator $A$ satisfies

$$P(P^\top(\Omega)) = P_F(\Omega), \quad \forall \Omega \in S^p_V, \tag{59}$$
$$P^\top(\Sigma) = 0.$$

The penalty function $g_*$ is the convex conjugate of the "log-det" barrier function on $\mathbb{S}^p_V$ :

$$g_*(S) = -\min_{\Omega \in \mathbb{S}^p_V} \{S \bullet \Omega - \log \det \Omega\}$$

Assuming that $V$ is chordal, the function $g_*(S)$, its gradient $\nabla g_*(S)$, and its Hessian matrix-vector product $\nabla^2 g_*(S)[Y]$ can all be evaluated in closed-form [62, 63, 52]; see also [64]. Furthermore, if the pattern is sparse, i.e. its number of elements in the pattern satisfies $|V| = O(p)$, then all of these operations can be performed to arbitrary accuracy in $O(p)$ time and memory.

It is standard to solve 58 using Newton's method. Starting from some initial point $y_1 \in \text{dom } g$

$$y_{k+1} = y_k + \alpha_k \Delta y_k \quad \Delta y_k \equiv -\nabla^2 l(y_k)^{-1} \nabla l(y_k),$$

in which the step-size $\alpha_k$ is determined by backtracking line search, selecting the first instance of the sequence $\{1, \rho, \rho^2, \rho^3, \dots\}$ that satisfies the Armijo-Goldstein condition

$$l(y + \alpha \Delta y) \le l(y) + \gamma \alpha \Delta y^\top \nabla l(y),$$

in which $\gamma \in (0, 0.5)$ and $\rho \in (0, 1)$. The function $g_*$ is strongly self-concordant, and $l$ inherits this property from $g_*$. Accordingly, classical analysis shows that we require at most

$$\left\lceil \frac{l(y_1) - l(\hat{y})}{0.05\gamma\rho} + \log\log(1/\epsilon) \right\rceil \approx O(1) \text{ Newton steps}$$

to an $\epsilon$-optimal point satisfying $l(y_k) - l(\hat{y}) \leq \epsilon$. The bound is very pessimistic, and in practice, no more than 20-30 Newton steps are ever needed for convergence.

The most computationally expensive of Newton's method is the solution of the Newton direction $\Delta y$ via the $m \times m$ system of equations

$$\nabla^2 l(y_k) \Delta y = -\nabla l(y_k). \tag{60}$$

The main result in this section is a proof that the condition number of $\nabla^2 l(y)$ is independent of the problem dimension $n$.

**Theorem 5.** *At any $y$ satisfying $l(y) \leq l(y_1)$ and $\nabla l(y)^\top (y - y_1) \leq \phi_{\max}$, the condition number $\kappa_l$ of the Hessian matrix $\nabla^2 l(y)$ is bound*

$$\kappa_l \leq 4\left(1 + \frac{\phi_{\max}^2 \lambda_{\max}(\Omega_0)}{\lambda_{\min}(\hat{\Omega})}\right)^2 \tag{61}$$

*where:*

- $\phi_{\max} = l(y_1) - l(\hat{y})$ *is the suboptimality of the initial point,*

- $\mathbf{A} = [\text{vec } A_1, \ldots, \text{vec } A_m]$ *is the vectorized version of the problem data,*

- $\Omega_0 = -\nabla f_*(S_0)$ *and* $S_0 = \Sigma - A(y_0)$ *are the initial primal-dual pair,*

- $\hat{\Omega} = -\nabla f_*(\hat{S})$ *and* $\hat{S} = \Sigma - A(\hat{y})$ *are the solution primal-dual pair.*

As a consequence, each Newton direction can be computed in $O\left(\log \epsilon^{-1}\right)$ iterations using conjugate gradients, over $O\left(\log \log \epsilon^{-1}\right)$ total Newton steps. The overall minimization problem is solved to $\epsilon$-accuracy in

$$O\left(\log \epsilon^{-1} \log \log \epsilon^{-1}\right) \approx O(1)\text{CG iterations.}$$

The leading constant here is dependent polynomially on the problem data and the quality of the initial point, but independent of the problem dimensions.

*Proof* To prove the first bound (61), we will instead prove

$$\frac{\lambda_{\max}(\Omega)}{\lambda_{\min}(\Omega)} \leq 2 + \frac{2\phi_{\max}^2 \lambda_{\max}(\Omega_0)}{\lambda_{\min}(\hat{\Omega})}, \tag{62}$$

which yields the desired condition number bound on $\nabla^2 g(y)$. Writing $\lambda_1 = \lambda_{\max}(\Omega)$ and $\lambda_n = \lambda_{\min}(\Omega)$, we have from the two lemmas above:

$$\phi_{\max} \geq \lambda_{\min}(\hat{\Omega})\left(\sqrt{\lambda_n^{-1}} - \sqrt{\lambda_1^{-1}}\right)^2 > 0,$$

$$2\phi_{\max} \geq \lambda_{\min}\left(\Omega_0^{-1}\right)\left(\sqrt{\lambda_1} - \sqrt{\lambda_n}\right)^2 > 0.$$

Multiplying the two upper-bounds and substituing $\lambda_{\min}\left(\Omega_0^{-1}\right) = 1/\lambda_{\max}(\Omega_0)$ yields

$$\frac{2\phi_{\max}^2 \lambda_{\max}(\Omega_0)}{\lambda_{\min}(\hat{\Omega})} \geq \left(\sqrt{\frac{\lambda_1}{\lambda_n}} - \sqrt{\frac{\lambda_n}{\lambda_1}}\right)^2 = \frac{\lambda_1}{\lambda_n} + \frac{\lambda_n}{\lambda_1} - 2$$

Finally, bounding $\lambda_n/\lambda_1 \geq 0$ yields (62).

