# OpenReview forum: "FasMe: Fast and Sample-efficient Meta Estimator for Precision Matrix Learning in Small Sample Settings"
_NeurIPS.cc/2024/Conference — NeurIPS 2024 poster_

### Official Review · Reviewer_nam2 · 2024-07-08

**Soundness:** 3
**Presentation:** 3
**Contribution:** 3
**Rating:** 6
**Confidence:** 3

**Summary:**

This paper proposes a meta-learning method for estimating the precision matrix on a new task with small data.
The proposed method uses common edges estimated from multiple auxiliary datasets as meta-knowledge. Then, it estimates the precision matrix on the new task, assuming its true edges contain all the estimated common edges (meta-knowledge). Some theoretical guarantees are also provided.
Experiments with synthetic and real-world datasets show the effectiveness and efficiency of the proposed method.

**Strengths:**

- The paper is generally well-written and easy to follow.
- Concrete algorithm and its theoretical guarantees are presented (but, I didn't read their proof).
- Strong performance in terms of both accuracy and efficiency in the experiments.

**Weaknesses:**

- As the authors stated in the Limitation section, the assumption of Eq. 8 might not be held in general machine learning tasks, although it fits well in biological domains.

**Questions:**

Please see the Weaknesses.

**Limitations:**

Yes

---

> ### Author Rebuttal · Authors · 2024-08-07
>
> ### Response to Weakness
> Thank you for your detailed review and for highlighting this crucial aspect of our research. As mentioned in Line 149-150: "The assumption has been widely adopted [28, 18, 30] and has proven feasible and applicable in the biological and genetic domains." We have validated this assumption through numerous real-world experiments across various biological datasets, detailed in Section 6.2 and Appendix B.2.
>
> In the Limitations Section, we acknowledge potential challenges in sparse and high-dimensional datasets from other domains, noting: "For other domains, if the edges are sparse and the data are high-dimensional, it may be difficult to recover the common structure from the data without some prior evidence." However, we would like to respectfully point out that this assumption has been successfully applied in some other fields as well. For instance, references [9, 10] (See global rebuttal for references) validate this assumption in industrial sensor and stock price datasets, respectively. Additionally, we have applied this assumption to a financial dataset, as described in Lines 566-575 of the appendix. The inherent characteristics in financial data allow for the adaptation of this assumption. For example, the relationships between ships and transport or between steel and the chemical industry (shown in Figure 6 in Appendix) are common across various economic datasets, showcasing our method's versatility.
>
> Moving forward, we plan to extend our work by relaxing the sub-Gaussian data assumption and further reducing the sample size requirements for training meta-knowledge. This will enhance the applicability of our approach across a broader range of fields. We greatly appreciate your insightful feedback and constructive suggestions. We have added the above discussion to the revised paper.

---

> > ### Comment · Reviewer_nam2 · 2024-08-12
> >
> > Thank you for your response. I'll keep the current rating.

---

### Official Review · Reviewer_gJdm · 2024-07-09

**Soundness:** 3
**Presentation:** 3
**Contribution:** 3
**Rating:** 7
**Confidence:** 3

**Summary:**

This paper introduces FasMe, a meta-learning approach for efficient precision matrix estimation in small sample settings. By leveraging meta-knowledge and maximum determinant matrix completion, FasMe reduces sample size requirements and improves computational efficiency. Experimental results show FasMe to be significantly faster and more accurate than existing methods, particularly in low-data environments.

**Strengths:**

1) Paper investigates a key issue in precision matrix estimation and proposes a reasonable method to address the problem.

2) Paper provides thorough theoretical and experimental analyses to justify the method’s ability to reduce the sample requirement and enhance learning efficiency.

3) Paper has good representation.

**Weaknesses:**

I have few doubts about the method and experiments presented in the article.

**Questions:**

Is the work primarily applicable to biological scenarios, or can it be widely used in other contexts as well?

**Limitations:**

nan

---

> ### Author Rebuttal · Authors · 2024-08-07
>
> ### Response to Question
>
> Thank you for your positive feedback and insightful question.
>
> Yes, our work is primarily applied to biological scenarios.  For one thing, high-dimensional, small-sample settings are more common in biological research, as exemplified by our case study on Cholangiocarcinoma at the beginning of the paper. For another thing, as stated in Line 149-150 of our paper, "The assumption has been widely adopted [28, 18, 30] and proved to be feasible and applicable in the biological and genetic domains [2]." This highlights the strong foundation and successful application of our work in these fields.
>
> However, our work can also be applied to other domains, although perhaps not as widely as in the biological field. In the appendix (Line 566-575), we demonstrate the application of our method in the financial domain, presenting experimental results that show promising performance. As we mentioned above, the assumptions and dependencies leveraged by our method are more commonly found and well-studied in biological data. Nevertheless, the inherent characteristics of financial data allow for its adaptation. For example, the relationships between ships and transport or between steel and the chemical industry (shown in Figure 6 in Appendix) are common across various economic datasets, showcasing our method's versatility.
>
> We believe this is a valuable point of discussion and have added the discussion in the main text of the revised paper to elaborate on these aspects further.
>
> Thank you again for your thoughtful question.

---

### Official Review · Reviewer_PtUj · 2024-07-10

**Soundness:** 3
**Presentation:** 3
**Contribution:** 3
**Rating:** 6
**Confidence:** 3

**Summary:**

The authors propose a method to estimate sparse precision matrices from few samples. Theoretical properties of the proposed method are studied, and experiments on synthetic and brain fMRI data are presented.

**Strengths:**

Strengths:
* The paper is overall well written, and fairly easy to follow and comprehend.
* Theoretical guarantees for sub-Gaussian distributed random variables are presented, and they seem to be novel contribution.
* The experiments on the synthetic dataset clearly demonstrate improvement over the relevant competing methods.

**Weaknesses:**

Weaknesses:
* Currently quantitative results are presented for synthetic data. It would be nice to see more quantitative evaluations on benchmark datasets.

**Questions:**

* How are the meta learning tasks linked to learning the new precision matrix in the theoretical part? It would be nice if the authors can elaborate on the assumptions for the learnability of the new matrix.
2. Why is N=0 in Eq. 13?

---

> ### Author Rebuttal · Authors · 2024-08-07
>
> We appreciate your kind feedback and perceptive questions.
>
> ### 1. Response to Weakness
>
> In addition to the synthetic datasets, we have conducted extensive experiments on real-world datasets. Specifically, we used the ChIP-Seq dataset from the ENCODE project and the fMRI dataset from the OpenfMRI project, as described in Section 6.2 (Page 9, Line 335-345) and Appendix B.2 (Page 13-14, Line 518-565). Additionally, we have referenced Appendix B.2 in Line 345.
>
> The datasets used in our work have been widely utilized as the benchmark datasets in other research papers in this domain. We list some papers as evidence:
>
> - ChIP-Seq dataset from the ENCODE project:
>   - Mitra et al. (2013) [1] introduced a Bayesian graphical model for inferring chromatin states from **ChIP-Seq data** on histone modifications. This approach identifies complex dependencies among various histone modifications, enabling a better understanding of their roles in gene regulation.
>   - Lundberg et al. (2015) [2] developed ChromNet, a computational method to infer the chromatin network from **ChIP-Seq datasets**, which identifies conditional dependencies among regulatory factors to discern direct from indirect interactions more effectively. This approach enables the analysis of large-scale ChIP-Seq data, revealing both known and previously unidentified interactions among regulatory factors.
>   - Ng et al. (2018) [3] developed a graphical model to visualize regulatory relationships between genome-wide transcription factor binding profiles from **ChIP-Seq datasets**, demonstrating an innovative approach to discern direct versus indirect transcription factor interactions and enhance the understanding of transcriptional regulation.
>   - Shu et al. (2021) [4]  introduced a computational framework utilizing neural networks for the inference and visualization of gene regulatory networks (GRNs) from single-cell RNA-sequencing data. Their model significantly enhances the accuracy of gene interaction analysis within and across various cell types, leveraging **ChIP-Seq datasets** to validate inferred GRNs and providing a robust method for biological data integration and interpretation.
>
> - fMRI dataset from OpenfMRI project:
>   - Ryali et al. (2012) [5] developed a method for estimating functional connectivity in fMRI datasets using stability selection-based sparse partial correlation with an elastic net penalty. This approach provides more accurate identification of functional connections, even with large **fMRI datasets**, enhancing the understanding of brain function and connectivity.
>   - Using **fMRI datasets**, Luo (2014) [6] proposes a hierarchical graphical model to effectively estimate large inverse covariance matrices, facilitating improved inference of functional brain networks and their hierarchical interactions. This model addresses the challenges of high dimensionality in brain data, offering a more accurate understanding of brain connectivity.
>   - Sulek (2017) [7] utilized graphical models with a lasso penalty to analyze functional magnetic resonance imaging **(fMRI) datasets,** focusing on brain activities during saccadic eye movement tasks. The study applied the graphical lasso method to construct sparse graphs that elucidate the connections between different brain regions, comparing these connections before and after specific tasks to understand changes in brain connectivity.
>   - Chung et al. (2021) [8] developed a robust graphical model that jointly estimates multiple precision matrices from **fMRI datasets**, facilitating a detailed analysis of brain networks by integrating regularized aggregation to account for individual variability and detect outliers, significantly enhancing the accuracy and reliability of connectivity assessments across subjects.
>
> The above references can be seen in the global rebuttal.
>
> We appreciate your suggestion and will ensure to highlight these real-world datasets more clearly in the revised version of our paper. Additionally, we will include the aforementioned references in the revised version as they provide significant evidence of the relevance and utilization of these benchmark datasets in the field. Thank you again for your valuable feedback.
>
> ### 2. Response to Q1
>
> We appreciate the opportunity to clarify this point. Due to the space limitation in the main text, we have provided a detailed theoretical explanation of the assumptions that elaborate on the link between the auxiliary tasks and the new task in Appendix F, specifically in Definition 2 (Page 21, Lines 748-759). Additionally, we have referenced this in the main text on lines 153-154. We hope this addresses your concerns and are grateful for your attention to detail.
>
> ### 3. Response to Q2
>
> We apologize for any confusion our notation may have caused. The $N=0$ in Eq. 13 should be read in conjunction with the subsequent $\forall(i, j) \in \mathcal{G}\_M$. To avoid any misunderstanding, we will revise this equation to $\forall(i, j) \in \mathcal{G}\_M, N_{ij}=0$. We appreciate your attention to detail and your helpful suggestion.

---

> > ### Comment · Reviewer_PtUj · 2024-08-13
> >
> > Thanks to the authors for the rebuttal, I am keeping the initial ratings.

---

### Author Rebuttal · Authors · 2024-08-07

## References
[1] Mitra R, Müller P, Liang S, et al. A Bayesian graphical model for chip-seq data on histone modifications[J]. Journal of the American Statistical Association, 2013, 108(501): 69-80.

[2] Lundberg S M, Tu W B, Raught B, et al. Learning the human chromatin network from all ENCODE ChIP-seq data[J]. bioRxiv, 2015: 023911.

[3] Ng F S L, Ruau D, Wernisch L, et al. A graphical model approach visualizes regulatory relationships between genome-wide transcription factor binding profiles[J]. Briefings in bioinformatics, 2018, 19(1): 162-173.

[4] Shu H, Zhou J, Lian Q, et al. Modeling gene regulatory networks using neural network architectures[J]. Nature Computational Science, 2021, 1(7): 491-501.

[5] Ryali S, Chen T, Supekar K, et al. Estimation of functional connectivity in fMRI data using stability selection-based sparse partial correlation with elastic net penalty[J]. NeuroImage, 2012, 59(4): 3852-3861.

[6] Luo X. A hierarchical graphical model for big inverse covariance estimation with an application to fmri[J]. arXiv preprint arXiv:1403.4698, 2014.

[7] Sulek T R. An application of graphical models to fMRI data using the lasso penalty[D]. University of Georgia, 2017.

[8] Chung J, Jackson B S, Mcdowell J E, et al. Joint estimation and regularized aggregation of brain network in FMRI data[J]. Journal of Neuroscience Methods, 2021, 364: 109374.

[9] Hara S, Washio T. Learning a common substructure of multiple graphical Gaussian models[J]. Neural Networks, 2013, 38: 23-38.

[10] Banerjee S, Ghosal S. Bayesian structure learning in graphical models[J]. Journal of Multivariate Analysis, 2015, 136: 147-162.

---

### Decision · Program_Chairs · 2024-09-25

**Decision:**

Accept (poster)

**Comment:**

This paper introduces a meta learning technique for precision matrix estimation, e.g. estimating the precision matrix of a dataset X' given a set of training datasets X1,...,Xk. Overall the reviewers appreciated the theoretical results for sub-Gaussian RVs and strong empirical efficiency improvements. There were no major concerns with the paper; I might simply recommend that the authors' make the results currently in Appendix B.2 more prominently visible.